# Support Recovery in Sparse PCA with General Missing Data

**Hanbyul Lee**[1]  **Qifan Song**[1]  **Jean Honorio**[2]

[1]Department of Statistics, Purdue University
[2]School of Computing and Information Systems, The University of Melbourne

## Abstract

We analyze a sparse PCA algorithm for incomplete and noisy data without any specific model assumption on the data missing scheme. We utilize a graphical approach to characterize general missing patterns, which enables us to analyze the effect of structural properties of missing patterns on the solvability of sparse PCA problem. The sparse PCA method we focus on is a semidefinite relaxation of the $\ell_1$-regularized PCA problem. We provide theoretical justification that the support of the sparse leading eigenvector can be recovered with high probability using the algorithm, under certain conditions. The conditions involve the spectral gap between the largest and second-largest eigenvalues of the true data matrix, the magnitude of the noise, and the structural properties of the missing pattern. The concepts of algebraic connectivity and irregularity are used to describe the properties in a graphical way. We empirically justify our theorem with synthetic data analysis. We show that the SDP algorithm outperforms other sparse PCA approaches especially when the observation pattern has good structural properties. As a by-product of our analysis, we provide two theorems to handle general missing schemes, which can be applied to other problems related to incomplete data matrices.

## 1 INTRODUCTION

When principal components possess a certain sparsity structure, standard principal component analysis (PCA) is not preferred due to poor interpretability and the inconsistency of solutions under high-dimensional settings [Paul, 2007, Nadler, 2008, Johnstone and Lu, 2009]. To solve these issues, *sparse PCA* has been proposed, which enforces sparsity in the PCA solution so that dimension reduction and variable selection can be simultaneously performed. Theoretical and algorithmic research on sparse PCA has been actively conducted over the past few years [Zou et al., 2006, Amini and Wainwright, 2008, Journée et al., 2010, Berk and Bertsimas, 2019, Richtárik et al., 2021].

In this paper, we focus on the case that the data to which sparse PCA is applied are not completely observed, but partially missing. Missing data frequently occurs in a wide range of machine learning problems, and sparse PCA is no exception. This has led to several works that offer reliable solutions to sparse PCA on missing data [Lounici, 2013, Kundu et al., 2015, Park and Zhao, 2019, Lee et al., 2022]. However, these methods make a restrictive assumption that the entries are observed according to a specific probabilistic (uniform) model.

Assuming a certain probabilistic model on the missing schemes can pose several problems. Firstly, in real-world scenarios, it is unfeasible to verify such assumptions beyond controlled experiments. Moreover, even though verification were feasible, the theorems under the assumptions could be lack of universality; they are only applicable in restrictive situations where the model assumption aligns. Furthermore, there is a risk of overlooking crucial factors that affect the problem's solvability but are hard to be captured within a specific model assumption on the missing scheme (e.g., certain structural properties of the missing pattern.)

Therefore, we aim to analyze incomplete data without any specific model assumptions on the missing patterns. The strategy we use is a graphical approach. Figure 1 demonstrates how to construct a graph when incomplete data is given. We first create a binary matrix corresponding to the missing pattern, where observed entries are denoted as 1, while missing entries are marked as 0. This binary matrix can be thought of as an adjacency matrix of a graph (we call this an observation graph.) Then we can mathematically characterize the missing pattern by using its graph properties and use them to analyze the impact of the missing pattern on the solvability of the problem.

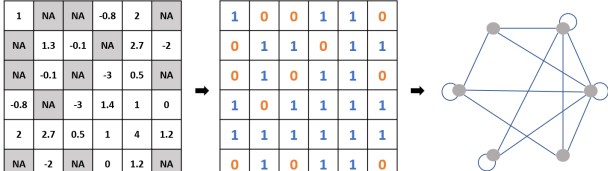

Figure 1: Example of incomplete matrix alongside its corresponding observation graph.

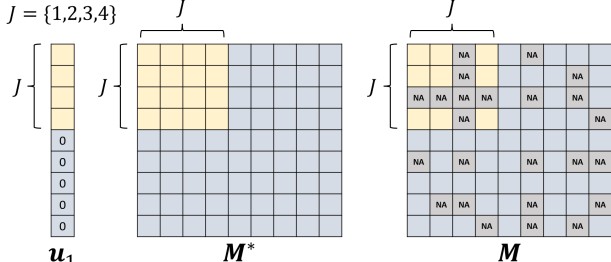

Figure 2: $\boldsymbol{u}_1$ represents a sparse leading eigenvector of the true complete matrix, $\boldsymbol{M}^*$, and $\boldsymbol{M}$ is the observed incomplete matrix. $\boldsymbol{u}_1$ contains non-zero entries in the yellow-colored portion. These non-zero entries only affect the entries in the yellow portion of $\boldsymbol{M}^*$. When this portion is not observed sufficiently evenly, we may fail to recover $J$. In this example, we may fail to identify index 3 as part of $J$.

This approach has multiple advantages. First, we can directly analyze given instances of missing patterns without presupposing any particular model for the missing scheme. Accordingly, there is no need to verify any model assumption, and our results can be universally applied. Additionally, we can use various graph properties to analyze structural properties of missing patterns that can impact the solvability of the problem. This is particularly important in sparse PCA, and the reason for this can be examined through a simple example.

Imagine that the leading eigenvector $\boldsymbol{u}$ of a symmetric matrix $\boldsymbol{M}^*$ is sparse and denote its support by $J$. In Figure 2, $J$ consists of the first four indices: 1, 2, 3, and 4, and the corresponding entries in $\boldsymbol{u}$ are highlighted in yellow. The goal of our sparse PCA problem is to exactly recover the support $J$ from an incomplete data matrix $\boldsymbol{M}$. Note that non-zero values in $\boldsymbol{u}$ only impact the entries in the $|J| \times |J|$ sub-matrix of $\boldsymbol{M}^*$, which is colored yellow in Figure 2 and comprises rows and columns indexed by $J$. Therefore, to recover the support $J$, we need to observe a sufficient number of entries in this sub-matrix. In fact, if we do not observe any entry in a specific row of the sub-matrix (e.g., the third row/column in $\boldsymbol{M}$ of the figure), we can never identify the corresponding index as an element of the support, and thus exact recovery fails. This implies that we need to observe the entries in the $|J| \times |J|$ sub-matrix abundantly and evenly; in other words, if we think of an observation graph, its sub-graph consisting of the nodes indexed by $J$ needs to be well-connected and have similar node degrees. These structural properties can be mathematically expressed by using graph properties that we will introduce later.

The sparse PCA algorithm we focus on is a semidefinite relaxation of the $l_1$-regularized PCA (we call this the SDP algorithm). This algorithm has been analyzed on complete data and has been shown to have theoretically good properties [d'Aspremont et al., 2004, Lei and Vu, 2015]. It has also been shown to work well for incomplete data, when the observation rate is sufficiently large under the uniform random sampling scheme [Lee et al., 2022]. In fact, the SDP algorithm works well under general missing schemes as well, especially when the missing pattern has good structural properties. In such cases, the SDP algorithm outperforms other sparse PCA methods. We will show this in a later section of the paper.

Our main contribution is as follows: we provide theoretical justification (i.e., Theorem 1) that when incomplete and noisy observations are given, the true support $J$ can be exactly recovered with high probability by using the SDP algorithm. Our theorem does not rely on any model assumption of the missing schemes. The sufficient condition we derive involves the spectral gap between the largest and second-largest eigenvalues of the true matrix, the magnitude of noise, and especially, structural properties of the observation graph. Two interpretable graph properties are involved: algebraic connectivity and irregularity (see Definitions 1 and 2). Through these graph properties, we demonstrate that the algorithm works well if the sub-graph consisting of the nodes indexed by $J$ is well-connected and has similar node degrees. It is important to note that the graph properties apply to any type of undirected graph. That is, our theorem is applicable to any given instances of missing pattern. To the best of our knowledge, this is the first work on sparse PCA with incomplete data without any specific model assumption on the missingness.

We empirically validate our theorem with synthetic data analysis in Section 5. Our simulation results show that the performance of the SDP algorithm is solely determined by the properties we derive in our theorem, which is a strong justification of our theory. We also show that the SDP algorithm outperforms several other sparse PCA approaches, and only the SDP algorithm benefits from the good structure of the observation graph.

As by-products of our analysis, we provide two theorems to handle the general missing schemes: the tail bound for the spectral norm of a random matrix having independent sub-Gaussian values over a fixed subset of entries, and the bound of the spectral norm of the difference between complete and incomplete matrices under general missingness (see Section 4.1). These theorems are important to prove our main theorem. Furthermore, they can be applied to other

problems related to incomplete data as well.

**Notation.** Matrices are bold capital (e.g., $\boldsymbol{A}$), vectors are bold lowercase (e.g., $\boldsymbol{a}$), and scalars or entries are not bold. $A_{i,j}$ and $a_i$ represent the $(i,j)$-th and $i$-th entries of $\boldsymbol{A}$ and $\boldsymbol{a}$, respectively. For any index sets $I$ and $J$, $\boldsymbol{A}_{I,J}$ and $\boldsymbol{a}_I$ denote the $|I| \times |J|$-dimensional sub-matrix of $\boldsymbol{A}$ consisting of rows in $I$ and columns in $J$, and the $|I|$-dimensional sub-vector of $\boldsymbol{a}$ consisting of the entries in $I$, respectively. For any positive integer $n$, we denote $[n] := \{1, \ldots, n\}$. $\|\boldsymbol{a}\|_1$ and $\|\boldsymbol{a}\|_2$ represent the $l_1$ and $l_2$ norms of $\boldsymbol{a}$. $\|\boldsymbol{A}\|_2$ and $\|\boldsymbol{A}\|_*$ indicate the spectral and nuclear norms of $\boldsymbol{A}$. We let $\|\boldsymbol{A}\|_{1,1} = \sum_i \sum_j |A_{i,j}|$ and $\|\boldsymbol{A}\|_{\max} = \max_{i,j} |A_{i,j}|$. $\lambda_i(\boldsymbol{A})$ represents the $i$-th largest eigenvalue of $\boldsymbol{A}$. The trace of $\boldsymbol{A}$ is denoted by $tr(\boldsymbol{A})$, and the matrix inner product of $\boldsymbol{A}$ and $\boldsymbol{B}$ is denoted by $\langle \boldsymbol{A}, \boldsymbol{B} \rangle$. $\boldsymbol{A} \circ \boldsymbol{B}$ represents the Hadamard product of $\boldsymbol{A}$ and $\boldsymbol{B}$. $f(x) = O(g(x))$ means that there exists a positive constant $C$ such that $f(x) \leq Cg(x)$ asymptotically. $f(x) = \Omega(g(x))$ is equivalent to $g(x) = O(f(x))$. $f(x) = \tilde{O}(g(x))$ is shorthand for $f(x) = O(g(x) \log^k x)$ for some $k > 0$.

## 2 PROBLEM DEFINITION

**Sparse Principal Component.** Let $\boldsymbol{M}^* \in \mathbb{R}^{d \times d}$ be an unknown symmetric matrix and $\boldsymbol{M}^* = \sum_{k \in [d]} \lambda_k(\boldsymbol{M}^*) \boldsymbol{u}_k \boldsymbol{u}_k^\top$ be the spectral decomposition of $\boldsymbol{M}^*$, where $\lambda_1(\boldsymbol{M}^*) > \lambda_2(\boldsymbol{M}^*) \geq \cdots \geq \lambda_d(\boldsymbol{M}^*)$ are its eigenvalues and $\boldsymbol{u}_1, \ldots, \boldsymbol{u}_d \in \mathbb{R}^d$ are the corresponding eigenvectors. For identifiability of the leading eigenvector, we consider that $\lambda_1(\boldsymbol{M}^*)$ is strictly greater than $\lambda_2(\boldsymbol{M}^*)$. In this paper, we assume strict sparsity in the leading eigenvector $\boldsymbol{u}_1$ of $\boldsymbol{M}^*$, as in previous studies [Amini and Wainwright, 2008, Vu et al., 2013, Gu et al., 2014, Wang et al., 2014, Lei and Vu, 2015, Deshp et al., 2016, Gataric et al., 2020, Agterberg and Sulam, 2022]. That is, we suppose that for some index set $J \in [d]$,

$$\begin{cases} u_{1,i} \neq 0 & \text{if } i \in J, \\ u_{1,i} = 0 & \text{otherwise.} \end{cases}$$

With a notation $supp(\boldsymbol{a}) := \{i \in [d] : a_i \neq 0\}$ for any vector $\boldsymbol{a} \in \mathbb{R}^d$, we can write $J = supp(\boldsymbol{u}_1)$. Also, we denote the size of $J$ by $s$. That is, $s = |J|$.

**Incomplete and Noisy Observation.** Suppose that we have only noisy observations of the entries of $\boldsymbol{M}^*$ over a fixed sampling set $\Omega \subseteq [d] \times [d]$. Specifically, we observe a symmetric matrix $\boldsymbol{M} \in \mathbb{R}^{d \times d}$ such that

$$M_{i,j} = \begin{cases} M_{i,j}^* + N_{i,j} & \text{if } (i,j) \in \Omega, \\ 0 & \text{otherwise} \end{cases}$$

for $i, j \in [d]$, where $N_{i,j}$ is the noise at location $(i,j)$. We assume that $N_{i,j}$'s are symmetric about zero and follow a sub-Gaussian distribution independently, i.e., $\mathbb{E} e^{\theta N_{i,j}} \leq e^{\frac{\sigma^2 \theta^2}{2}}$ for any $\theta \geq 0$ and some $\sigma \geq 0$.

**Goal.** In this paper, we aim to exactly recover the true support $J$ of the leading eigenvector $\boldsymbol{u}_1$ of $\boldsymbol{M}^*$ from the incomplete and noisy observation $\boldsymbol{M}$.

## 3 METHODS

In this section, we introduce the algorithm used to solve the sparse PCA problem, and define several graph properties which will be utilized in our main theorem.

**SDP Algorithm.** For the support recovery of the leading eigenvector, an intuitive approach is imposing a regularization term on the PCA quadratic loss. When using the $l_1$ regularizer, the optimization problem can be written as:

$$\hat{\boldsymbol{x}} = \underset{\|\boldsymbol{x}\|_2 = 1}{\arg \max} \, \boldsymbol{x}^\top \boldsymbol{M} \boldsymbol{x} - \rho \|\boldsymbol{x}\|_1^2.$$

Here, the true support $J$ is estimated with $supp(\hat{\boldsymbol{x}})$. However, the objective is non-convex and difficult to solve. Therefore, the following semidefinite relaxation is considered as an alternative:

$$\hat{\boldsymbol{X}} = \underset{\boldsymbol{X} \succeq 0 \text{ and } tr(\boldsymbol{X}) = 1}{\arg \max} \langle \boldsymbol{M}, \boldsymbol{X} \rangle - \rho \|\boldsymbol{X}\|_{1,1}. \quad (1)$$

In this paper, we call this the SDP algorithm. By letting $\boldsymbol{X} = \boldsymbol{x} \boldsymbol{x}^\top$, the equivalence of the above two objective functions can be easily justified. Since $supp(\boldsymbol{x}) = supp(diag(\boldsymbol{x} \boldsymbol{x}^\top))$, $J$ can be estimated by $\hat{J} = supp(diag(\hat{\boldsymbol{X}}))$. Efficient scalable SDP solvers exist [Yurtsever et al., 2021], so the SDP algorithm is computationally friendly. This approach has been shown to have good theoretical properties and work well in practice for both of complete and uniformly random missing data [d'Aspremont et al., 2004, Lei and Vu, 2015, Lee et al., 2022].

*Remark* 1. We note that in the implementation of the SDP algorithm, we use the matrix $\boldsymbol{M}$ where zero is imputed in the missing entries, without applying any matrix completion or imputation methods. Given that SDP's objective function is linear, imputing zeros for missing entries implies the utilization of information solely from observed entries during the optimization process. We note that matrix completion can introduce unwanted bias under inappropriate conditions. It is well-known that most of the matrix completion methods can be successful only under the low-rank assumption. In this paper, we allow the true matrix $\boldsymbol{M}^*$ to be not necessarily low-rank. In Section 5.2, we provide experimental evidence showing that the SDP algorithm with zero-imputed $\boldsymbol{M}$ performs well when the observation has a good structural property, while the result yielded from matrix completion does not achieve good performance overall. Imputing zeros is also beneficial in certain application fields. For example,

in the single-cell RNA sequence (scRNA-seq) data analysis, missing occurs due to the so-called dropout events, and it is hard to differentiate between true zeros, where the genes are not expressed at all, and the dropout zeros. We do not need to differentiate them when we impute zeros for missing entries.

**Graph Properties.** Before presenting our main theorem in the next section, we define several graph terminologies and properties which are involved in the theorem.

We first introduce the observation graph $\mathcal{G} = (\mathcal{V}, \mathcal{E})$, which is an undirected graph associated with the given sampling set $\Omega$, that is, $\mathcal{V} = [d]$ and $(i,j) \in \mathcal{E}$ if and only if $(i,j) \in \Omega$. Note that $\mathcal{G}$ is allowed to contain loops. We denote the adjacency matrix corresponding to $\mathcal{G}$ by $\boldsymbol{A}_{\mathcal{G}}$. With this notation, we can write $\boldsymbol{M} = \boldsymbol{A}_{\mathcal{G}} \circ (\boldsymbol{M}^* + \boldsymbol{N})$ where $\boldsymbol{N}$ is the noise matrix whose $(i,j)$-th entry is $N_{i,j}$.

Below are several convenient notations about graphs.

- $\mathcal{G}_{J,J}, \mathcal{G}_{J,J^c}, \mathcal{G}_{J^c,J^c}$: For the observation graph $\mathcal{G}$, we denote by $\mathcal{G}_{J,J}, \mathcal{G}_{J,J^c}$ and $\mathcal{G}_{J^c,J^c}$ the sub-graphs of $\mathcal{G}$ which consist of only the edges inside $J \times J$, $J \times J^c$, and $J^c \times J^c$, respectively. $\mathcal{G}_{J,J}$ and $\mathcal{G}_{J^c,J^c}$ are undirected graphs with vertex sets $J$ and $J^c$, respectively. $\mathcal{G}_{J,J^c}$ is a bipartite graph with independent vertex sets $J$ and $J^c$.
- $\overline{\mathcal{G}}$: For any graph $\mathcal{G}$, $\overline{\mathcal{G}}$ denotes it complement graph, i.e., $\overline{\mathcal{G}}$ has the same vertex set as $\mathcal{G}$ but its edge set is the complement of that of $\mathcal{G}$.
- $\Delta_{\max}(\mathcal{G}), \Delta_{\min}(\mathcal{G})$: For any graph $\mathcal{G}$, $\Delta_{\max}(\mathcal{G})$ and $\Delta_{\min}(\mathcal{G})$ denote the maximum and minimum node degrees of $\mathcal{G}$, respectively.

Now, we define two important structural graph properties, *algebraic connectivity* and *irregularity*. Both properties are crucial to explain the effect of the structure of the observation graph on the solvability of our support recovery problem.

Algebraic connectivity, the well-known concept to measure the graph connectivity, is defined as follows:

**Definition 1** (Algebraic Connectivity). The algebraic connectivity of a graph $\mathcal{G}$, denoted by $\phi(\mathcal{G})$, is the second-smallest eigenvalue of the Laplacian matrix of $\mathcal{G}$. The magnitude of $\phi(\mathcal{G})$ reflects **how well connected the overall graph is**.

Next, we introduce a new graphical property that we name 'irregularity', which proves crucial in presenting our results.

**Definition 2** (Irregularity). For any undirected graph $\mathcal{G}$ such that $\Delta_{\max}(\mathcal{G}) \geq \phi(\mathcal{G})$ and $\Delta_{\max}(\overline{\mathcal{G}}) \geq \phi(\overline{\mathcal{G}})$, the irregularity of $\mathcal{G}$ is defined as

$$\psi(\mathcal{G}) := \max\left\{ \Delta_{\max}(\mathcal{G}) - \phi(\mathcal{G}), \Delta_{\max}(\overline{\mathcal{G}}) - \phi(\overline{\mathcal{G}}) \right\}.$$

The magnitude of $\psi(\mathcal{G})$ reflects **how different the node degrees of $\mathcal{G}$ are**.

To better interpret the concept of irregularity, we derive some lower and upper bound results as below:

$$\max_{\|\boldsymbol{x}\|_2=1, \boldsymbol{x} \perp \mathbf{1}} \boldsymbol{x}^\top \boldsymbol{A}_{\mathcal{G}} \boldsymbol{x} \leq \Delta_{\max}(\mathcal{G}) - \phi(\mathcal{G})$$
$$\leq \max_{\|\boldsymbol{x}\|_2=1, \boldsymbol{x} \perp \mathbf{1}} \boldsymbol{x}^\top \boldsymbol{A}_{\mathcal{G}} \boldsymbol{x} + \Delta_{\max}(\mathcal{G}) - \Delta_{\min}(\mathcal{G})$$

where $\mathbf{1} = (1, 1, \ldots, 1)^\top$.

- $\max_{\|\boldsymbol{x}\|_2=1, \boldsymbol{x} \perp \mathbf{1}} \boldsymbol{x}^\top \boldsymbol{A}_{\mathcal{G}} \boldsymbol{x}$: Among different $\boldsymbol{A}_{\mathcal{G}}$'s having the same largest and second-largest eigenvalues, we can see that the one corresponding to a regular graph (a graph is *regular* when each node has the same degree) has the smallest magnitude of $\max_{\|\boldsymbol{x}\|_2=1, \boldsymbol{x} \perp \mathbf{1}} \boldsymbol{x}^\top \boldsymbol{A}_{\mathcal{G}} \boldsymbol{x}$. This is because a regular graph has a normalized vector of $\mathbf{1}$ as its leading eigenvector.
- $\Delta_{\max}(\mathcal{G}) - \Delta_{\min}(\mathcal{G})$: This quantity decreases as nodes of $\mathcal{G}$ have similar degrees.

Hence, we can say that as $\mathcal{G}$ becomes closer to a regular graph, the value of $\Delta_{\max}(\mathcal{G}) - \phi(\mathcal{G})$ decreases. Also, as $\mathcal{G}$ approaches a regular graph, $\overline{\mathcal{G}}$ also tends to be closer to a regular graph, resulting in a decrease in the value of $\Delta_{\max}(\overline{\mathcal{G}}) - \phi(\overline{\mathcal{G}})$. Consequently, as $\mathcal{G}$ approaches a regular graph, the value of $\psi(\mathcal{G}) = \max\left\{ \Delta_{\max}(\mathcal{G}) - \phi(\mathcal{G}), \Delta_{\max}(\overline{\mathcal{G}}) - \phi(\overline{\mathcal{G}}) \right\}$ decreases. This is the reason why we name this concept the 'irregularity'.

*Remark* 2. Note that $\Delta_{\max}(\mathcal{G}) \geq \phi(\mathcal{G})$ and $\Delta_{\max}(\overline{\mathcal{G}}) \geq \phi(\overline{\mathcal{G}})$, i.e., $\psi(\mathcal{G}) \geq 0$ holds except for the case that $\mathcal{G}$ or $\overline{\mathcal{G}}$ is a complete graph without loops.

## 4  MAIN RESULTS

Now, we introduce our main theorem, which shows the sufficient condition for the SDP algorithm to exactly recover the true support $J$. Because the sufficient condition involves multiple interrelated factors, parsing the effects of each component might be difficult. Hence, following the presentation of the main theorem, we will break down the sufficient condition into several segments for better understanding of the impact of each factor.

**Theorem 1.** *Under the problem definition in Section 2, assume that with some constant $c > 0$,*

$$\|\boldsymbol{M}^*_{J,J}\|_2 \cdot \psi(\mathcal{G}_{J,J}) + \sigma\sqrt{\Delta_{\max}(\mathcal{G}_{J,J})\log(s\log d)}$$
$$+ s\|\boldsymbol{M}^*_{J^c,J}\|_2 + \frac{1}{\sqrt{s}}\|\boldsymbol{M}^*_{J^c,J^c}\|_2$$
$$+ \sigma s\sqrt{\max\left\{\Delta_{\max}(\mathcal{G}_{J,J^c}), \Delta_{\max}(\mathcal{G}_{J^c,J^c})\right\}\log d}$$
$$\leq \frac{c\phi(\mathcal{G}_{J,J})\bar{\lambda}(\boldsymbol{M}^*) \cdot \min_{i \in J}|u_{1,i}|}{s} \quad (2)$$

where $\bar{\lambda}(\boldsymbol{M}^*) := \lambda_1(\boldsymbol{M}^*) - \lambda_2(\boldsymbol{M}^*)$ *and* $\psi(\mathcal{G}_{J,J}) \geq 0$. *Then, the optimal solution* $\hat{\boldsymbol{X}}$ *to the optimization problem* (1) *is unique and satisfies* $supp(diag(\hat{\boldsymbol{X}})) = J$ *with probability at least* $1 - O(s^{-1})$, *where* $\rho = 2\sigma\sqrt{\max\left\{\Delta_{\max}(\mathcal{G}_{J,J^c}), \Delta_{\max}(\mathcal{G}_{J^c,J^c})\right\}\log d} + \|\boldsymbol{M}^*_{J^c,J}\|_{\max}$.

In a nutshell, Theorem 1 asserts that the SDP algorithm produces a reliable solution under certain conditions imposed on the spectral gap $\bar{\lambda}(\boldsymbol{M}^*)$, the noise intensity parameter $\sigma$, the matrix norms, and the graph properties of the sub-graphs $\mathcal{G}_{J,J}$, $\mathcal{G}_{J,J^c}$ and $\mathcal{G}_{J^c,J^c}$.

To better understand the condition (2), we consider the setting that $\|\boldsymbol{M}^*_{J,J}\|_2 = O(\bar{\lambda}(\boldsymbol{M}^*))$ and $\min_{i \in J} |u_{1,i}| = \Omega(s^{-\frac{1}{2}})$, for instance. We note that the first inequality holds as long as $\|\boldsymbol{M}^*_{J,J}\|_2 = \lambda_1(\boldsymbol{M}^*_{J,J})$ and $\frac{\lambda_2(\boldsymbol{M}^*)}{\lambda_1(\boldsymbol{M}^*)} \leq c$ for some $c < 1$, and the second inequality holds when all the non-zero entries of the sparse leading eigenvector are of the same level of magnitude. In this case, we can rewrite (2) as follows:

$$\frac{\psi(\mathcal{G}_{J,J})}{\phi(\mathcal{G}_{J,J})} = O\left(\frac{1}{s\sqrt{s}}\right),$$

$$\sigma = \tilde{O}\left(\frac{\phi(\mathcal{G}_{J,J})}{\sqrt{\max\{\Delta_{\max}(\mathcal{G}_{J,J^c}), \Delta_{\max}(\mathcal{G}_{J^c,J^c})\}}} \cdot \frac{\bar{\lambda}(\boldsymbol{M}^*)}{s^2\sqrt{s}}\right),$$

$$\|\boldsymbol{M}^*_{J^c,J}\|_2 = O\left(\frac{\phi(\mathcal{G}_{J,J})\bar{\lambda}(\boldsymbol{M}^*)}{s^2\sqrt{s}}\right),$$

$$\|\boldsymbol{M}^*_{J^c,J^c}\|_2 = O\left(\frac{\phi(\mathcal{G}_{J,J})\bar{\lambda}(\boldsymbol{M}^*)}{s}\right).$$

The first condition about the structural graph properties of $\mathcal{G}_{J,J}$ states that for the algorithm to be successful, the sub-graph $\mathcal{G}_{J,J}$ is desired to have sufficiently large connectivity $\phi(\mathcal{G}_{J,J})$ and small irregularity $\psi(\mathcal{G}_{J,J})$. This implies that the sub-graph $\mathcal{G}_{J,J}$ needs to be well-connected and have similar node degrees, i.e., we need to observe the entries in the corresponding sub-matrix of the true matrix abundantly and evenly. This result fits well with our first intuition discussed in the introduction.

The other conditions mean that the noise and the norms of $\boldsymbol{M}^*_{J^c,J}$ and $\boldsymbol{M}^*_{J^c,J^c}$ need to be well-controlled for the success of the algorithm. This is in accordance with our common sense, since large $\sigma$ or $\boldsymbol{M}^*$ values outside $J \times J$ matrix will mask the true information. These conditions are alleviated when the connectivity of $\mathcal{G}_{J,J}$ is large (especially in the second condition, larger than the number of observed entries outside $J \times J$) and when the spectral gap $\bar{\lambda}(\boldsymbol{M}^*)$ is large.

We note that a sufficiently large spectral gap requirement is to ensure the uniqueness and identifiability of the projection matrix with respect to the principal subspace.

*Remark* 3. It is worth mentioning that the SDP method offers more than just support recovery; under the sufficient condition in Theorem 1, the optimal solution $\hat{\boldsymbol{X}}$ can be represented as $\hat{\boldsymbol{X}} = \begin{pmatrix} \hat{\boldsymbol{x}}\hat{\boldsymbol{x}}^\top & 0 \\ 0 & 0 \end{pmatrix}$, where $\hat{\boldsymbol{x}}$ satisfies $\left\|\boldsymbol{u}_1 - \begin{pmatrix} \hat{\boldsymbol{x}} \\ \boldsymbol{0} \end{pmatrix}\right\|_2$ or $\left\|\boldsymbol{u}_1 + \begin{pmatrix} \hat{\boldsymbol{x}} \\ \boldsymbol{0} \end{pmatrix}\right\|_2 \leq \min_{j \in J} |u_{1,j}|$. Given that $\min_{j \in J} |u_{1,j}| = O(s^{-\frac{1}{2}})$, the $\ell_2$ estimation error bound of the leading eigenvector $\boldsymbol{u}_1$ has a rate of $O(s^{-\frac{1}{2}})$.

*Remark* 4. When the target matrix $\boldsymbol{M}^*$ is a covariance matrix, and the algorithm (1) is applied to an incomplete sample covariance matrix, we can derive a result similar to Theorem 1, with $\sigma$ replaced by a function of the sample size. Suppose that $\boldsymbol{Z}_1, \ldots, \boldsymbol{Z}_n$ are drawn from the multivariate normal distribution $N(\boldsymbol{0}, \boldsymbol{M}^*)$, and consider an incomplete sample covariance matrix $\boldsymbol{M} = \boldsymbol{A}_{\mathcal{G}} \circ (\frac{1}{n} \sum_{i \in [n]} \boldsymbol{Z}_i \boldsymbol{Z}_i^\top)$. If $n > d$, we have the same result as in Theorem 1 where $\sigma$ is replaced by $\|\boldsymbol{M}^*\|_2 \cdot \frac{\sqrt{d \log d} \log n}{\sqrt{n}}$. This result can be derived by applying Lemma 6 instead of Lemma 5 in the proof.

*Remark* 5. Our work primarily focuses on the theoretical understanding of the problem and the algorithm, but we briefly discuss a practical use of our results here. Our results present that as we observe the entries well in the sub-matrix $\boldsymbol{M}^*_{J,J}$, we can recover the support $J$. Unfortunately, we do not know the true support $J$ in practice, so we cannot check if the observation structure satisfies the sufficient condition. One alternative is to check if any sub-graphs of the observation graph have good algebraic connectivity and irregularity. If there is a sub-graph with connectivity or irregularity that is extremely low or high, then we could conservatively suspect that the result from the algorithm cannot be fully trusted. Here, prior knowledge about the size of the support will be needed.

**Challenge in the Proof of Theorem 1**   Detailed proof of Theorem 1 is given in Section 7. At a high level, we use the KKT conditions under the primal-dual witness framework, which is standard in support recovery problems. However, simply applying this framework to the problem does not yield meaningful results. The main challenge lies in handling missingness without any model assumption on it. To circumvent this challenge, we employ observation graphs and graphical concepts. More importantly, we obtain and utilize two theorems: one is for the tail bound of the matrix whose entries in a fixed subset are random. This is to obtain a bound for sub-Gaussian noise only existing on the fixed sampling set of entries, $\Omega$. Also, the other is for bounding the difference between complete and incomplete matrices under a fixed sampling scheme. These two results can be widely used in other matrix-related problems with general missing data. We introduce these two theorems as by-products of our analysis in the next section.

### 4.1 BY-PRODUCTS

Below is a tail bound for the spectral norm of a random matrix with independent sub-Gaussian values in its fixed subset of entries.

**Theorem 2** (Tail Bound for Random Matrix with Independent Sub-Gaussian Values in a Fixed Subset of Entries). *Consider a random matrix $\boldsymbol{Z} \in \mathbb{R}^{m \times n}$ whose entries in a fixed subset independently follow a sub-Gaussian distribution which is symmetric about zero and has parameter $\sigma > 0$, while the other entries are fixed as zero. That is, there exists an index set $S \subseteq \{(i,j) \mid i \in [m], j \in [n]\}$ such that for $i \in [m]$ and $j \in [n]$,*

$$Z_{i,j} = \begin{cases} N_{i,j} & if\ (i,j) \in S, \\ 0 & otherwise, \end{cases}$$

*where each $N_{i,j}$ is symmetric about zero and satisfies $\mathbb{E}e^{\theta N_{i,j}} \leq e^{\frac{\sigma^2 \theta^2}{2}}$ for any $\theta > 0$. Then,*

$$\|\boldsymbol{Z}\|_2 \leq 2\sigma \sqrt{\Delta_{\max}(\mathcal{G}_S) \log(m+n)}$$

*with probability at least $1 - 2(m+n)^{-1}$, where $\mathcal{G}_S$ is a bipartite graph whose vertex sets are $[m]$ and $[n]$, and edge set is $S$.*

We defer the proof to Section 8. In the derivation of Theorem 1, we use the above theorem to obtain the tail bounds of $\|(\boldsymbol{A}_{\mathcal{G}})_{J,J} \circ \boldsymbol{N}_{J,J}\|_2$, $\|(\boldsymbol{A}_{\mathcal{G}})_{J,J^c} \circ \boldsymbol{N}_{J,J^c}\|_2$ and $\|(\boldsymbol{A}_{\mathcal{G}})_{J^c,J^c} \circ \boldsymbol{N}_{J^c,J^c}\|_2$, where $\boldsymbol{N}$ is the noise matrix whose entries follow a sub-Gaussian distribution independently.

*Remark* 6. When $N_{i,j}$'s follow independent normal distributions with mean 0 and variance $\sigma^2$, one can obtain a tighter concentration inequality by using Corollary 3.11 in Bandeira and Van Handel [2016]. For some positive constant $c$,

$$\|\boldsymbol{Z}\|_2 \leq c\sigma\big\{\sqrt{\Delta_{\max}(\mathcal{G}_S)} + \sqrt{\log(\min\{m,n\})}\big\}$$

holds with probability at least $1 - \min\{m,n\}^{-1}$.

Next is the bound of the spectral norm of the difference between complete and incomplete matrices under fixed missingness. We note that this is an extended result of Theorem 4.1 in Bhojanapalli and Jain [2014]. While the theorem of Bhojanapalli and Jain [2014] is limited to the case that the observation graph is regular, our theorem applies to any general undirected observation graph, so our result generalizes the result of Bhojanapalli and Jain [2014]. For regular graphs, our bound coincides with that of Bhojanapalli and Jain [2014].

**Theorem 3.** *Consider a symmetric matrix $\boldsymbol{Y}$ with dimension $n$. Let $\boldsymbol{Y} = \sum_{k \in [r]} \lambda_k(\boldsymbol{Y}) \boldsymbol{v}_k \boldsymbol{v}_k^\top$ be the spectral decomposition of $\boldsymbol{Y}$, where $r$ is rank of $\boldsymbol{Y}$. Define $\tau := \max_{i \in [n]} \sum_{k \in [r]} v_{k,i}^2$.*

*Also, consider an undirected graph $\mathcal{G}$ with $n$ nodes and denote its adjacency matrix by $\boldsymbol{A}_{\mathcal{G}}$. Then,*

$$\left\|\boldsymbol{Y} - \frac{n}{\phi(\mathcal{G})}\boldsymbol{A}_{\mathcal{G}} \circ \boldsymbol{Y}\right\|_2 \leq \frac{n\tau\psi(\mathcal{G})}{\phi(\mathcal{G})} \cdot \|\boldsymbol{Y}\|_2.$$

The proof is given in Section 9. The main challenge in the proof is to find and use proper graph properties to make the bound of the difference small enough. We use the above theorem to bound $\|\boldsymbol{M}^*_{J,J} - \frac{s}{\phi(\mathcal{G}_{J,J})}(\boldsymbol{A}_{\mathcal{G}})_{J,J} \circ \boldsymbol{M}^*_{J,J}\|_2$ in the proof of Theorem 1.

### 4.2 CHOICE OF THE TUNING PARAMETER

The theoretical choice of $\rho$ in Theorem 1 is useless in practice since it relies on unknown quantities. Therefore, certain tuning procedure over $\rho$ is necessary for the implementation of the SDP algorithm (1). Our suggestion is to find $\rho$ to maximize the following AIC type criterion (see also Qi et al. [2013]):

$$C_\rho = (1-a)\frac{\langle \boldsymbol{M}, \hat{\boldsymbol{X}}_\rho \rangle}{\langle \boldsymbol{M}, \hat{\boldsymbol{X}}_0 \rangle} + a\Big(1 - \frac{|supp(diag(\hat{\boldsymbol{X}}_\rho))|}{d}\Big).$$

Here, $\hat{\boldsymbol{X}}_\rho$ and $\hat{\boldsymbol{X}}_0$ refer to the solutions of the SDP algorithm where the tuning parameters are set to be $\rho$ and 0, respectively. $\langle \boldsymbol{M}, \hat{\boldsymbol{X}}_\rho \rangle$ and $\langle \boldsymbol{M}, \hat{\boldsymbol{X}}_0 \rangle$ represent the explained variances of $\boldsymbol{M}$ by the solutions. The first term of the criterion is a measure for the quality of the estimate, and the second term penalizes for the complexity of the solution. $a \in (0,1)$ is the weight to be chosen by practitioners. As one needs a sparse principal component, a relatively large value of $a$ is suggested. In the experiments, we find that $0.4 \leq a \leq 0.6$ generally work well.

## 5 EXPERIMENTS

### 5.1 SYNTHETIC DATA ANALYSIS

The goal of this synthetic data analysis is to demonstrate the effects of the structural properties of the observation graph, the spectral gap between the largest and second-largest eigenvalues of the true matrix, and the magnitude of the noise, on the success of the support recovery by the SDP algorithm.

In particular, we check whether the performance of the SDP algorithm (1) is solely determined by the properties we derive in Theorem 1. We utilize the following rescaled parameter for this:

$$Rescaled = \frac{\text{LHS of (2)}}{\text{RHS of (2) without constant } c}. \quad (3)$$

If the performance of the algorithm versus this rescaled parameter is the same across different settings, then we can

empirically justify that the performance is solely determined by the factors in the rescaled parameter. This kind of approach has been used in Wainwright [2009] for sparse linear regression.

**Setting.** We use synthetic data generated in the following manner. The orthonormal eigenvectors of $\boldsymbol{M}^*$ are randomly selected, while the leading eigenvector $\boldsymbol{u}_1$ is made to be sparse and have $s$ non-zero entries with the values of $\frac{1}{\sqrt{s}}$. $\lambda_2(\boldsymbol{M}^*), \ldots, \lambda_d(\boldsymbol{M}^*)$ are randomly selected from a normal distribution with mean 0 and standard deviation 1, and $\lambda_1(\boldsymbol{M}^*)$ is set to $\lambda_2(\boldsymbol{M}^*)$ plus the spectral gap. We set the matrix dimension $d$ to be 50 and the support size $s$ to be 10.

We generate the observation graph $\mathcal{G}$ to have 1250 edges out of 2500. We set the values of $\frac{\psi(\mathcal{G}_{J,J})}{\phi(\mathcal{G}_{J,J})}$ to be included in one of the ranges 0 to 2, 2 to 4, ..., or 16 to 18. Recall that $\phi(\mathcal{G}_{J,J})$ measures how well connected the sub-graph $\mathcal{G}_{J,J}$ is, and $\psi(\mathcal{G}_{J,J})$ measures how different the node degrees of $\mathcal{G}_{J,J}$ are. Therefore, as $\frac{\psi(\mathcal{G}_{J,J})}{\phi(\mathcal{G}_{J,J})}$ decreases, $\mathcal{G}_{J,J}$ is well-connected and has similar node degrees, that is, the entries of the corresponding $|J| \times |J|$ sub-matrix are observed more abundantly and evenly.

The entry-wise noise is randomly selected from a normal distribution with mean 0 and standard deviation $\sigma$. In each setting, we run the algorithm (1) and examine if the solution exactly recovers the true support $J$. The tuning parameter $\rho \in \{0.025, 0.5, \ldots, 1\}$ is selected by the method in Section 4.2 with $a = 0.5$. We repeat each experiment 100 times with different random seeds, and calculate the rate of exact recovery in each setting.

**Results.** In Figure 3, plots (a) and (b) show the experimental results where we fix the noise parameter $\sigma$ as 0 (noiseless) and try different spectral gaps $\bar{\lambda}(\boldsymbol{M}^*) \in \{1, 2, 5, 10\}$ to check the effect of the spectral gap. Plots (c) and (d) show the results where we fix the spectral gap $\bar{\lambda}(\boldsymbol{M}^*)$ as 20 and try different noise parameters $\sigma \in \{0.1, 0.3, 0.5, 0.7\}$ to check the effect of the magnitude of noise. From (a), we can observe that the exact recovery rate increases as the spectral gap increases, which is consistent with our theoretical finding. From (c), we can check that the exact recovery rate increases as the standard deviation of the noise decreases, which also supports our theorem. In (a) and (c), it is shown that as the value of $\frac{\psi(\mathcal{G}_{J,J})}{\phi(\mathcal{G}_{J,J})}$ decreases - indicating that the entries of $|J| \times |J|$ sub-matrix are observed more abundantly and evenly - the exact recovery rate increases. This aligns with our initial intuition.

Lastly, in (b) and (d), we can see that the curves of the exact recovery rate versus the rescaled parameter share almost the same pattern under different settings of $\bar{\lambda}(\boldsymbol{M}^*)$ and $\sigma$. This provides empirical justification of our theorem in the sense that the performance of the SDP algorithm is solely determined by the properties we derive in Theorem 1.

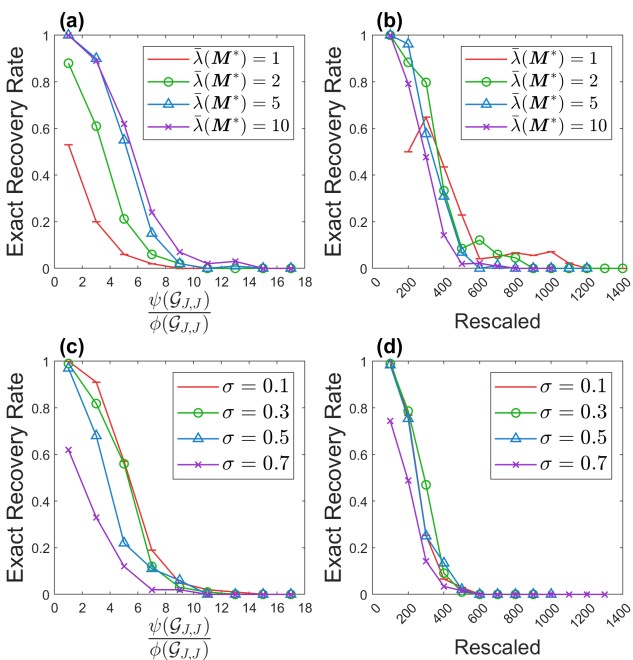

Figure 3: (a) Rate of exact recovery of $J$ versus $\frac{\psi(\mathcal{G}_{J,J})}{\phi(\mathcal{G}_{J,J})}$ for different $\bar{\lambda}(\boldsymbol{M}^*)$ where $\sigma = 0$. (b) Same simulation results as in (a) with exact recovery rate plotted versus rescaled parameter (3). (c) Rate of exact recovery of $J$ versus $\frac{\psi(\mathcal{G}_{J,J})}{\phi(\mathcal{G}_{J,J})}$ for different $\sigma$ where $\bar{\lambda}(\boldsymbol{M}^*) = 20$. (d) Same simulation results as in (c) with exact recovery rate plotted versus rescaled parameter (3).

We provide additional simulation results for cases where the density of the observation graph varies in Appendix 11, for interested readers.

## 5.2 SEMI-SYNTHETIC DATA ANALYSIS

The pitprops data [Jeffers, 1967], which stores 180 observations of 13 variables, has been a standard benchmark to evaluate algorithms for sparse PCA (see, e.g., Zou et al. [2006], Shen and Huang [2008], Journée et al. [2010], Qi et al. [2013]). It has been revealed that on the complete pitprops data, a sparse solution with 6 nonzero entries (with respect to the variables 'topdiam', 'length', 'ringbut', 'bowmax', 'bowdist', 'whorls') has a comparable explained variance with that of the dense solution from original PCA. By generating observation graphs synthetically, we construct semi-synthetic incomplete matrices of the pitprops data and aim to recover the 6 nonzero entries with incomplete data.

The primary goal of this experimental study is to check if the SDP method performs well compared to other sparse PCA methods on incomplete data. By demonstrating the superior performance of the SDP method over others, we further support our proposal of applying the SDP method to incomplete data. We will also validate the selection criterion

of the tuning parameter $\rho$ in Section 4.2.

**Setting.** We impose missingness and noise on the complete covariance matrix in the following manner. We generate the observation graph $\mathcal{G}$ to have 100 edges out of 169. The value of $\frac{\psi(\mathcal{G}_{J,J})}{\phi(\mathcal{G}_{J,J})}$ is set to be included in one of the ranges 0 to 0.2, 0.2 to 0.4, ..., or 2 to 2.2. The entry-wise noise is randomly selected from a normal distribution with mean 0 and standard deviation $\sigma = 0.1$. On average, the ratio of the spectral norm of the generated noise matrix to the spectral norm of the noiseless matrix is about 0.153.

We compare the SDP algorithm with seven different methods. First, we consider two well-known sparse PCA algorithms and one recently-proposed method: the generalized power method (GPM) by Journée et al. [2010], the iterative thresholding sparse PCA (ITSPCA) by Ma [2013] and the alternating manifold proximal gradient method (A-ManPG) by Chen et al. [2020]. All methods have theoretically good properties on complete data, and we checked that they succeed in support recovery on the complete pitprops data with proper hyper-parameter choices. We implement these methods on incomplete data with zero imputation for missing cells. Second, we consider the combination of matrix completion and each of the SDP, GPM, ITSPCA and A-ManPG algorithms. We estimate the missing entries of the incomplete matrix $\boldsymbol{M}$ by using the following matrix completion method based on the nuclear norm minimization:

$$\tilde{\boldsymbol{M}} = \underset{\boldsymbol{Y}=\boldsymbol{Y}^\top, \; \boldsymbol{A}_\mathcal{G} \circ \boldsymbol{Y} = \boldsymbol{A}_\mathcal{G} \circ \boldsymbol{M}}{\arg\min} \|\boldsymbol{Y}\|_*.$$

Then we implemented the SDP, GPM, ITSPCA and A-ManPG algorithms with the completed matrix $\tilde{\boldsymbol{M}}$.

We run each algorithm 100 times with different random seeds in each setting, and calculate the rate of exact recovery of $J$. In the SDP algorithm, the tuning parameter $\rho \in \{0.025, 0.5, \ldots, 1\}$ is selected by the method in Section 4.2 with $a = 0.4$. For the tuning parameter of the GPM algortihm, we consider $\gamma \in \{0.2, 0.4, \ldots, 2\} \cup \{\max_{i \in [d]} \|\boldsymbol{M}_{i,:}\|_2\}$ as suggested by Journée et al. [2010]. For the ITSPCA and A-ManPG algorithms, there are no well-known methods to choose tuning parameters, so we try multiple values of tuning parameters and choose ones with the largest exact recovery rates.

**Results.** In Figure 4, we can see that our method of selecting $\rho$ works well. Here, we compare the result from our tuning method with those in the settings where the tuning parameter $\rho$ is fixed as a value among $\{0.1, 0.2, \ldots, 1\}$ through all the repetitions. We can see that the exact recovery rate from our tuning method is larger than most of the results where $\rho$ is fixed as one value.

Figure 5 shows that the SDP method outperforms the other sparse PCA methods when the observation graph has a good structural property. First, we observe that all of the GPM,

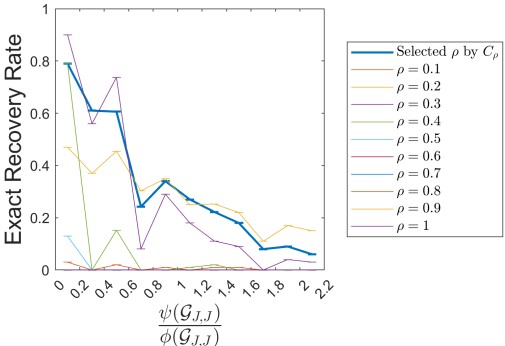

Figure 4: Rate of exact recovery of $J$ versus $\frac{\psi(\mathcal{G}_{J,J})}{\phi(\mathcal{G}_{J,J})}$ for SDP algorithm with different values of $\rho$. Thick blue line indicates result of $\rho$ which is selected by criterion $C_\rho$.

ITSPCA and A-ManPG algorithms perform worse than the SDP algorithm, especially in scenarios with favorable observation graph properties, characterized by small values of $\frac{\psi(\mathcal{G}_{J,J})}{\phi(\mathcal{G}_{J,J})}$. Unlike the SDP method, these algorithms cannot benefit from good structure of the observation graph. In addition, the SDP method with matrix completion has the exact recovery rate of around 0.4 overall, while the SDP algorithm with zero-imputation produces the exact recovery rate greater than 0.6 when the value of $\frac{\psi(\mathcal{G}_{J,J})}{\phi(\mathcal{G}_{J,J})}$ is small enough. Also, the GPM, ITSPCA and A-ManPG algorithms with matrix completion exhibit similar or worse performance compared to the SDP algorithm with matrix completion. That is, any methods with matrix completion do not perform better than the SDP algorithm with zero-imputation.

About the failure of the matrix completion approach, we conjecture the following rationale: while the matrix completion algorithm can be successful under the low-rank assumption, the pitprops data is full-rank with eigenvalues ranging from 0.039 to 4.219, which yields an unsuccessful matrix completion result. The average of relative error $\frac{\|\tilde{\boldsymbol{M}} - \boldsymbol{M}^*\|_F}{\|\boldsymbol{M}^*\|_F}$ for the matrix completion solution $\tilde{\boldsymbol{M}}$ was 0.504, with no instance showing an error smaller than 0.2. Accordingly, the imputed cells introduce more noise into the inference and the result of sparse PCA becomes even worse than that of simply using zero for the missing entries. We note that unlike matrix completion, the SDP algorithm with zero-imputation does not require the low-rank assumption to be successful according to our theorem. Therefore, the algorithm has good performance under a general condition on the true matrix.

# 6 CONCLUDING REMARKS

This paper examines the support recovery problem in sparse PCA with incomplete and noisy data, under a general sampling scheme. We consider a practical algorithm based on a semidefinite relaxation of the $\ell_1$-regularized PCA problem, and provide sufficient conditions where the algorithm can

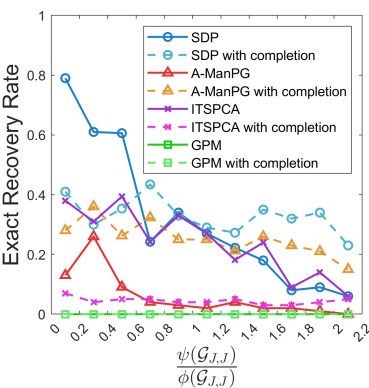

Figure 5: Rate of exact recovery of $J$ versus $\frac{\psi(\mathcal{G}_{J,J})}{\phi(\mathcal{G}_{J,J})}$ for eight different sparse PCA algorithms.

exactly recover the true support with high probability. The conditions involve the spectral gap, the noise parameter, the matrix norms, and the structural properties of the observation graph. We show that the algorithm works well if we observe the entries in the sub-matrix $\boldsymbol{M}^*_{J,J}$ abundantly and evenly. We empirically justify our theorem with synthetic data analysis, and show that the SDP algorithm outperforms several other sparse PCA approaches especially when the observation graph has a good structural property.

For clarity of exposition, this paper has focused on the case where only the leading eigenvector of $\boldsymbol{M}^*$ is sparse. We briefly discuss how to extend the framework to the case where multiple leading eigenvectors of $\boldsymbol{M}^*$ are sparse. Let us assume that $r$ leading eigenvectors of $\boldsymbol{M}^*$ are sparse. In this case, the constraint $tr(\boldsymbol{X}) = 1$ in (1) should be replaced with a constraint $tr(\boldsymbol{X}) = r$. To derive the sufficient conditions to exactly recover the true (joint) support $J$ for several leading eigenvectors, we could make use of the primal-dual witness framework in a similar way as done in Theorem 1 for one leading eigenvector. We leave this for future research.

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

# Support Recovery in Sparse PCA with General Missing Data
## (Supplementary Materials)

**Hanbyul Lee**[1]  **Qifan Song**[1]  **Jean Honorio**[2]

[1]Department of Statistics, Purdue University
[2]School of Computing and Information Systems, The University of Melbourne

## 7  PROOF OF THEOREM 1

**Step 1: Deriving Sufficient Conditions From KKT Conditions (Primal-dual Witness Approach)**

With the primal variable $\boldsymbol{X} \in \mathbb{R}^{d \times d}$ and the dual variables $\boldsymbol{Z} \in \mathbb{R}^{d \times d}$, $\boldsymbol{\Lambda} \in \mathbb{R}^{d \times d}$ and $\mu \in \mathbb{R}$, the Lagrangian of the problem (1) is written as

$$L(\boldsymbol{X}, \boldsymbol{Z}, \boldsymbol{\Lambda}, \mu) = -\langle \boldsymbol{M}, \boldsymbol{X} \rangle + \rho \langle \boldsymbol{X}, \boldsymbol{Z} \rangle - \langle \boldsymbol{\Lambda}, \boldsymbol{X} \rangle + \mu \cdot (tr(\boldsymbol{X}) - 1)$$

where $Z_{ij} \in \partial |X_{ij}|$ for each $i, j \in [d]$. According to the standard KKT condition, we can derive that $(\hat{\boldsymbol{X}}, \hat{\boldsymbol{Z}}, \hat{\boldsymbol{\Lambda}}, \hat{\mu})$ is optimal if and only if the followings hold:

- Primal feasibility: $\hat{\boldsymbol{X}} \succeq 0, tr(\hat{\boldsymbol{X}}) = 1$
- Dual feasibility: $\hat{\boldsymbol{\Lambda}} \succeq 0, \hat{Z}_{ij} \in \partial |\hat{X}_{ij}|$ for each $i, j \in [d]$
- Complementary slackness: $\langle \hat{\boldsymbol{\Lambda}}, \hat{\boldsymbol{X}} \rangle = 0$ ($\Leftrightarrow \hat{\boldsymbol{\Lambda}} \hat{\boldsymbol{X}} = 0$ if $\hat{\boldsymbol{X}} \succeq 0$ and $\hat{\boldsymbol{\Lambda}} \succeq 0$)
- Stationarity: $\hat{\boldsymbol{\Lambda}} = -\boldsymbol{M} + \rho \hat{\boldsymbol{Z}} + \hat{\mu} \cdot \boldsymbol{I}$.

By substituting $\hat{\boldsymbol{\Lambda}}$ with $-\boldsymbol{M} + \rho \hat{\boldsymbol{Z}} + \hat{\mu} \cdot \boldsymbol{I}$, it can be shown that the above conditions are equivalent to

$$\hat{\boldsymbol{X}} \succeq 0, tr(\hat{\boldsymbol{X}}) = 1$$
$$\boldsymbol{M} - \rho \hat{\boldsymbol{Z}} \preceq \hat{\mu} \boldsymbol{I}$$
$$\hat{Z}_{ij} \in \partial |\hat{X}_{ij}| \quad \text{for each } i, j \in [d]$$
$$(\boldsymbol{M} - \rho \hat{\boldsymbol{Z}}) \hat{\boldsymbol{X}} = \hat{\mu} \cdot \hat{\boldsymbol{X}}.$$

To use the primal-dual witness construction, we now consider the following restricted problem:

$$\max_{\boldsymbol{X} \succeq 0, tr(\boldsymbol{X})=1 \text{ and } supp(\boldsymbol{X}) \subseteq J \times J} \langle \boldsymbol{M}, \boldsymbol{X} \rangle - \rho \|\boldsymbol{X}\|_{1,1}. \tag{4}$$

Similarly to the above, we can derive that $\hat{\boldsymbol{X}} = \begin{pmatrix} \hat{\boldsymbol{X}}_{J,J} & 0 \\ 0 & 0 \end{pmatrix}$[1] is optimal to the problem (4) if and only if

$$\hat{\boldsymbol{X}}_{J,J} \succeq 0, tr(\hat{\boldsymbol{X}}_{J,J}) = 1$$
$$\boldsymbol{M}_{J,J} - \rho \hat{\boldsymbol{Z}}_{J,J} \preceq \hat{\mu} \boldsymbol{I}$$
$$\hat{Z}_{ij} \in \partial |\hat{X}_{ij}| \quad \text{for each } i, j \in J$$
$$(\boldsymbol{M}_{J,J} - \rho \hat{\boldsymbol{Z}}_{J,J}) \hat{\boldsymbol{X}}_{J,J} = \hat{\mu} \cdot \hat{\boldsymbol{X}}_{J,J}.$$

Now, we want for the above solution $\hat{\boldsymbol{X}} = \begin{pmatrix} \hat{\boldsymbol{X}}_{J,J} & 0 \\ 0 & 0 \end{pmatrix}$ to satisfy the optimality conditions of the original problem (1).

Furthermore, by assuming the strict dual feasibility, we want to guarantee $supp(diag(\hat{\boldsymbol{X}})) \subseteq J$. We can easily derive the sufficient conditions listed below:

$$\hat{\boldsymbol{X}}_{J,J} \succeq 0, tr(\hat{\boldsymbol{X}}_{J,J}) = 1$$
$$\boldsymbol{M}_{J,J} - \rho\hat{\boldsymbol{Z}}_{J,J} \preceq \hat{\mu}\boldsymbol{I}$$
$$\boldsymbol{M} - \rho\hat{\boldsymbol{Z}} \preceq \hat{\mu}\boldsymbol{I}$$
$$\hat{Z}_{ij} \in \partial|\hat{X}_{ij}| \quad \text{for each } (i,j) \in J \times J$$
$$\hat{Z}_{ij} \in (-1, 1) \quad \text{for each } (i,j) \notin J \times J$$
$$(\boldsymbol{M}_{J,J} - \rho\hat{\boldsymbol{Z}}_{J,J})\hat{\boldsymbol{X}}_{J,J} = \hat{\mu} \cdot \hat{\boldsymbol{X}}_{J,J}$$
$$(\boldsymbol{M}_{J^c,J} - \rho\hat{\boldsymbol{Z}}_{J^c,J})\hat{\boldsymbol{X}}_{J,J} = 0.$$

If the above conditions hold, then $\hat{\boldsymbol{X}} = \begin{pmatrix} \hat{\boldsymbol{X}}_{J,J} & 0 \\ 0 & 0 \end{pmatrix}$ is optimal to the problem (1) and satisfies $supp(diag(\hat{\boldsymbol{X}})) \subseteq J$.

Now, consider $\hat{\boldsymbol{x}}, \hat{\boldsymbol{z}} \in \mathbb{R}^s$ such that

$$\hat{z}_i = \text{sign}(u_{1,i}) \quad \text{for all } i \in J,$$
$$\hat{\boldsymbol{x}} \text{ is the leading eigenvector of } \boldsymbol{M}_{J,J} - \rho\hat{\boldsymbol{z}}\hat{\boldsymbol{z}}^\top. \tag{5}$$

Let $\hat{\boldsymbol{X}}_{J,J} = \hat{\boldsymbol{x}}\hat{\boldsymbol{x}}^\top$ and $\hat{\boldsymbol{Z}}_{J,J} = \hat{\boldsymbol{z}}\hat{\boldsymbol{z}}^\top$. Then if the following conditions hold:

$$\text{sign}(u_{1,i}) = \text{sign}(\hat{x}_i) \text{ for all } i \in J \quad \text{or} \quad \text{sign}(u_{1,i}) = -\text{sign}(\hat{x}_i) \text{ for all } i \in J \tag{6}$$
$$(\boldsymbol{M}_{J^c,J} - \rho\hat{\boldsymbol{Z}}_{J^c,J})\hat{\boldsymbol{x}} = 0 \quad \text{and} \quad \|\hat{\boldsymbol{Z}}_{J^c,J}\|_{\max} < 1 \tag{7}$$
$$\lambda_1(\boldsymbol{M}_{J,J} - \rho\hat{\boldsymbol{z}}\hat{\boldsymbol{z}}^\top) = \lambda_1(\boldsymbol{M} - \rho\hat{\boldsymbol{Z}}) \quad \text{and} \quad \|\hat{\boldsymbol{Z}}_{J^c,J^c}\|_{\max} < 1, \tag{8}$$

the above sufficient conditions are satisfied, that is, $\hat{\boldsymbol{X}} := \begin{pmatrix} \hat{\boldsymbol{x}}\hat{\boldsymbol{x}}^\top & 0 \\ 0 & 0 \end{pmatrix}$ is optimal to the problem (1). Also, $supp(diag(\hat{\boldsymbol{X}})) = J$ holds since $\text{sign}(u_{1,i}) = \text{sign}(\hat{x}_i \text{ or } -\hat{x}_i) \neq 0$ for all $i \in J$.

For the uniqueness, we need an additional condition presented in the following lemma.

**Lemma 1.** *For $\hat{\boldsymbol{X}}$ and $\hat{\boldsymbol{Z}}$ constructed above, if the following condition holds:*

$$\lambda_1(\boldsymbol{M}_{J,J} - \rho\hat{\boldsymbol{z}}\hat{\boldsymbol{z}}^\top) > \lambda_2(\boldsymbol{M}_{J,J} - \rho\hat{\boldsymbol{z}}\hat{\boldsymbol{z}}^\top) \tag{9}$$

*then the solution $\hat{\boldsymbol{X}}$ is a unique optimal solution to the problem (1).*

*Proof.* According to the standard primal-dual witness construction, we only need to show that under the condition, $\hat{\boldsymbol{X}}_{J,J} = \hat{\boldsymbol{x}}\hat{\boldsymbol{x}}^\top$ is a unique optimal solution to the restricted problem (4).

Assume that there exists another optimal solution to the problem (4), say $\tilde{\boldsymbol{X}}_{J,J}$. Also, denote its dual optimal solution by $\tilde{\boldsymbol{Z}}_{J,J}$. Then, we can write

$$\langle \boldsymbol{M}_{J,J}, \hat{\boldsymbol{X}}_{J,J} \rangle - \rho\|\hat{\boldsymbol{X}}_{J,J}\|_{1,1} = \langle \boldsymbol{M}_{J,J} - \rho\hat{\boldsymbol{z}}\hat{\boldsymbol{z}}^\top, \hat{\boldsymbol{x}}\hat{\boldsymbol{x}}^\top \rangle = \hat{\boldsymbol{x}}^\top(\boldsymbol{M}_{J,J} - \rho\hat{\boldsymbol{z}}\hat{\boldsymbol{z}}^\top)\hat{\boldsymbol{x}}$$
$$= \langle \boldsymbol{M}_{J,J}, \tilde{\boldsymbol{X}}_{J,J} \rangle - \rho\|\tilde{\boldsymbol{X}}_{J,J}\|_{1,1} = \langle \boldsymbol{M}_{J,J} - \rho\tilde{\boldsymbol{Z}}_{J,J}, \tilde{\boldsymbol{X}}_{J,J} \rangle.$$

Recall that $\hat{\boldsymbol{x}}$ is the leading eigenvector of $\boldsymbol{M}_{J,J} - \rho\hat{\boldsymbol{z}}\hat{\boldsymbol{z}}^\top$, that is, $\hat{\boldsymbol{x}}^\top(\boldsymbol{M}_{J,J} - \rho\hat{\boldsymbol{z}}\hat{\boldsymbol{z}}^\top)\hat{\boldsymbol{x}} = \lambda_1(\boldsymbol{M}_{J,J} - \rho\hat{\boldsymbol{z}}\hat{\boldsymbol{z}}^\top)$. Now, we will show that $\langle \boldsymbol{M}_{J,J} - \rho\hat{\boldsymbol{z}}\hat{\boldsymbol{z}}^\top, \tilde{\boldsymbol{X}}_{J,J} \rangle < \lambda_1(\boldsymbol{M}_{J,J} - \rho\hat{\boldsymbol{z}}\hat{\boldsymbol{z}}^\top)$ for any matrix $\tilde{\boldsymbol{X}}_{J,J} \neq \hat{\boldsymbol{x}}\hat{\boldsymbol{x}}^\top$ such that $\tilde{\boldsymbol{X}}_{J,J} \succeq 0$ and $tr(\tilde{\boldsymbol{X}}_{J,J}) = 1$.

---

[1]For clarity of exposition, our abuse of notation seemingly assumes $J = [s]$ when we join vectors and matrices. It should be clear that for $J \neq [s]$, one will need to properly interleave vector entries or matrix rows/columns.

Let $\tilde{\boldsymbol{X}}_{J,J} = \sum_{i \in J} \theta_i \boldsymbol{v}_i \boldsymbol{v}_i^\top$, which is the spectral decomposition of $\tilde{\boldsymbol{X}}_{J,J}$. We can derive that

$$\langle \boldsymbol{M}_{J,J} - \rho \hat{\boldsymbol{z}}\hat{\boldsymbol{z}}^\top, \tilde{\boldsymbol{X}}_{J,J} \rangle = \langle \boldsymbol{M}_{J,J} - \rho \hat{\boldsymbol{z}}\hat{\boldsymbol{z}}^\top, \sum_{i \in J} \theta_i \boldsymbol{v}_i \boldsymbol{v}_i^\top \rangle = \sum_{i \in J} \theta_i \boldsymbol{v}_i^\top (\boldsymbol{M}_{J,J} - \rho \hat{\boldsymbol{z}}\hat{\boldsymbol{z}}^\top) \boldsymbol{v}_i \le \lambda_1 (\boldsymbol{M}_{J,J} - \rho \hat{\boldsymbol{z}}\hat{\boldsymbol{z}}^\top)$$

where the last inequality holds since $\sum_{i \in J} \theta_i = tr(\tilde{\boldsymbol{X}}_{J,J}) = 1$ and $\boldsymbol{v}_i^\top (\boldsymbol{M}_{J,J} - \rho \hat{\boldsymbol{z}}\hat{\boldsymbol{z}}^\top) \boldsymbol{v}_i \le \lambda_1 (\boldsymbol{M}_{J,J} - \rho \hat{\boldsymbol{z}}\hat{\boldsymbol{z}}^\top)$. Here, the equality holds only if $\theta_1 = 1$, $\theta_i = 0$ for $i \ne 1$ and $\boldsymbol{v}_1 = \hat{\boldsymbol{x}}$, that is, $\tilde{\boldsymbol{X}}_{J,J} = \hat{\boldsymbol{x}}\hat{\boldsymbol{x}}^\top$. Therefore, $\langle \boldsymbol{M}_{J,J} - \rho \hat{\boldsymbol{z}}\hat{\boldsymbol{z}}^\top, \tilde{\boldsymbol{X}}_{J,J} \rangle < \lambda_1 (\boldsymbol{M}_{J,J} - \rho \hat{\boldsymbol{z}}\hat{\boldsymbol{z}}^\top)$ for any matrix $\tilde{\boldsymbol{X}}_{J,J} \ne \hat{\boldsymbol{x}}\hat{\boldsymbol{x}}^\top$ such that $\tilde{\boldsymbol{X}}_{J,J} \succeq 0$ and $tr(\tilde{\boldsymbol{X}}_{J,J}) = 1$.

With this fact, we can derive that

$$\begin{aligned}
\langle \boldsymbol{M}_{J,J}, \hat{\boldsymbol{X}}_{J,J} \rangle - \rho \|\hat{\boldsymbol{X}}_{J,J}\|_{1,1} &= \hat{\boldsymbol{x}}^\top (\boldsymbol{M}_{J,J} - \rho \hat{\boldsymbol{z}}\hat{\boldsymbol{z}}^\top)\hat{\boldsymbol{x}} = \lambda_1 (\boldsymbol{M}_{J,J} - \rho \hat{\boldsymbol{z}}\hat{\boldsymbol{z}}^\top) \\
&> \langle \boldsymbol{M}_{J,J} - \rho \hat{\boldsymbol{z}}\hat{\boldsymbol{z}}^\top, \tilde{\boldsymbol{X}}_{J,J} \rangle = \langle \boldsymbol{M}_{J,J} - \rho \tilde{\boldsymbol{Z}}_{J,J}, \tilde{\boldsymbol{X}}_{J,J} \rangle + \rho \langle \tilde{\boldsymbol{Z}}_{J,J} - \hat{\boldsymbol{z}}\hat{\boldsymbol{z}}^\top, \tilde{\boldsymbol{X}}_{J,J} \rangle \\
&= \langle \boldsymbol{M}_{J,J}, \tilde{\boldsymbol{X}}_{J,J} \rangle - \rho \|\tilde{\boldsymbol{X}}_{J,J}\|_{1,1} + \rho \langle \tilde{\boldsymbol{Z}}_{J,J} - \hat{\boldsymbol{z}}\hat{\boldsymbol{z}}^\top, \tilde{\boldsymbol{X}}_{J,J} \rangle.
\end{aligned}$$

Since $\langle \boldsymbol{M}_{J,J}, \hat{\boldsymbol{X}}_{J,J} \rangle - \rho \|\hat{\boldsymbol{X}}_{J,J}\|_{1,1} = \langle \boldsymbol{M}_{J,J}, \tilde{\boldsymbol{X}}_{J,J} \rangle - \rho \|\tilde{\boldsymbol{X}}_{J,J}\|_{1,1}$ by the assumption, the above inequality implies $\langle \tilde{\boldsymbol{Z}}_{J,J} - \hat{\boldsymbol{z}}\hat{\boldsymbol{z}}^\top, \tilde{\boldsymbol{X}}_{J,J} \rangle < 0$, that is, $\langle \tilde{\boldsymbol{Z}}_{J,J}, \tilde{\boldsymbol{X}}_{J,J} \rangle < \langle \hat{\boldsymbol{z}}\hat{\boldsymbol{z}}^\top, \tilde{\boldsymbol{X}}_{J,J} \rangle$. This contradicts the fact that $\langle \tilde{\boldsymbol{Z}}_{J,J}, \tilde{\boldsymbol{X}}_{J,J} \rangle = \sup_{\|\boldsymbol{Z}_{J,J}\|_{\max} \le 1} \langle \boldsymbol{Z}_{J,J}, \tilde{\boldsymbol{X}}_{J,J} \rangle$, and thus the desired result holds.

$\square$

**Step 2: Deriving Sufficient Conditions for (6)-(9)**

**Lemma 2** (Sufficient Condition for (6)). *If the following inequality holds:*

$$\|\boldsymbol{M}_{J,J}^*\|_2 \cdot \psi(\mathcal{G}_{J,J}) + 2\sigma \sqrt{\Delta_{\max}(\mathcal{G}_{J,J}) \log s} + s\rho \le \frac{\phi(\mathcal{G}_{J,J})\bar{\lambda}(\boldsymbol{M}_{J,J}^*) \cdot \min_{i \in J} |u_{1,i}|}{2\sqrt{2}s},$$

*then the condition (6) holds, that is, $\mathrm{sign}(u_{1,i}) = \mathrm{sign}(\hat{x}_i)$ for all $i \in J$ or $\mathrm{sign}(u_{1,i}) = -\mathrm{sign}(\hat{x}_i)$ for all $i \in J$, with probability at least $1 - 2s^{-1}$.*

*Proof.* By applying the Davis-Kahan sin$\Theta$ theorem, we obtain

$$\|\boldsymbol{u}_1 - \hat{\boldsymbol{x}}\|_2 \text{ or } \|\boldsymbol{u}_1 + \hat{\boldsymbol{x}}\|_2 \le \frac{2\sqrt{2}}{\bar{\lambda}(\boldsymbol{M}_{J,J}^*)} \cdot \|\boldsymbol{M}_{J,J}^* - \frac{s}{\phi(\mathcal{G}_{J,J})}(\boldsymbol{M}_{J,J} - \rho \hat{\boldsymbol{z}}\hat{\boldsymbol{z}}^\top)\|_2.$$

By the triangle inequality, Lemma 5 and Theorem 3, we can upper bound

$$\begin{aligned}
\|\boldsymbol{M}_{J,J}^* - \frac{s}{\phi(\mathcal{G}_{J,J})}(\boldsymbol{M}_{J,J} - \rho \hat{\boldsymbol{z}}\hat{\boldsymbol{z}}^\top)\|_2 &\le \|\boldsymbol{M}_{J,J}^* - \frac{s}{\phi(\mathcal{G}_{J,J})}\mathbb{E}[\boldsymbol{M}_{J,J}]\|_2 + \frac{s}{\phi(\mathcal{G}_{J,J})}\|\mathbb{E}[\boldsymbol{M}_{J,J}] - \boldsymbol{M}_{J,J}\|_2 + \frac{s^2\rho}{\phi(\mathcal{G}_{J,J})} \\
&\le \|\boldsymbol{M}_{J,J}^* - \frac{s}{\phi(\mathcal{G}_{J,J})}(\boldsymbol{A}_\mathcal{G})_{J,J} \circ \boldsymbol{M}_{J,J}^*\|_2 + \frac{s}{\phi(\mathcal{G}_{J,J})} \cdot 2\sigma \sqrt{\Delta_{\max}(\mathcal{G}_{J,J}) \log s} + \frac{s^2\rho}{\phi(\mathcal{G}_{J,J})} \\
&\le \frac{s\psi(\mathcal{G}_{J,J})}{\phi(\mathcal{G}_{J,J})} \cdot \|\boldsymbol{M}_{J,J}^*\|_2 + \frac{s}{\phi(\mathcal{G}_{J,J})} \cdot 2\sigma \sqrt{\Delta_{\max}(\mathcal{G}_{J,J}) \log s} + \frac{s^2\rho}{\phi(\mathcal{G}_{J,J})}
\end{aligned}$$

with probability at least $1 - 2s^{-1}$.

Now, we have that

$$\|\boldsymbol{u}_1 - \hat{\boldsymbol{x}}\|_2 \text{ or } \|\boldsymbol{u}_1 + \hat{\boldsymbol{x}}\|_2 \le \frac{2\sqrt{2}}{\bar{\lambda}(\boldsymbol{M}_{J,J}^*)} \cdot \left\{ \frac{s\psi(\mathcal{G}_{J,J})}{\phi(\mathcal{G}_{J,J})} \cdot \|\boldsymbol{M}_{J,J}^*\|_2 + \frac{s}{\phi(\mathcal{G}_{J,J})} \cdot 2\sigma \sqrt{\Delta_{\max}(\mathcal{G}_{J,J}) \log s} + \frac{s^2\rho}{\phi(\mathcal{G}_{J,J})} \right\}.$$

By Lemma 7, if

$$\frac{2\sqrt{2}}{\bar{\lambda}(\boldsymbol{M}_{J,J}^*)} \cdot \left\{ \frac{s\psi(\mathcal{G}_{J,J})}{\phi(\mathcal{G}_{J,J})} \cdot \|\boldsymbol{M}_{J,J}^*\|_2 + \frac{s}{\phi(\mathcal{G}_{J,J})} \cdot 2\sigma \sqrt{\Delta_{\max}(\mathcal{G}_{J,J}) \log s} + \frac{s^2\rho}{\phi(\mathcal{G}_{J,J})} \right\} \le \min_{i \in J} |u_{1,i}|,$$

that is,

$$\|\boldsymbol{M}^*_{J,J}\|_2 \cdot \psi(\mathcal{G}_{J,J}) + 2\sigma\sqrt{\Delta_{\max}(\mathcal{G}_{J,J})\log s} + s\rho \leq \frac{\phi(\mathcal{G}_{J,J})\bar{\lambda}(\boldsymbol{M}^*_{J,J}) \cdot \min_{i\in J}|u_{1,i}|}{2\sqrt{2}s},$$

then $sign(u_{1,i}) = sign(\hat{x}_i)$ for all $i \in J$ or $sign(u_{1,i}) = -sign(\hat{x}_i)$ for all $i \in J$ with probability at least $1 - 2s^{-1}$.

$\square$

**Lemma 3** (Sufficient Condition for (7)). *Let* $\hat{\boldsymbol{Z}}_{J^c,J} = \frac{1}{\rho\|\hat{\boldsymbol{x}}\|_1}\boldsymbol{M}_{J^c,J}\hat{\boldsymbol{x}}\hat{\boldsymbol{z}}^\top$. *Then it satisfies* $(\boldsymbol{M}_{J^c,J} - \rho\hat{\boldsymbol{Z}}_{J^c,J})\hat{\boldsymbol{x}} = 0$. *Also, if the following inequality holds:*

$$2\sigma\sqrt{\Delta_{\max}(\mathcal{G}_{J,J^c})\log d} + \|\boldsymbol{M}^*_{J^c,J}\|_{\max} < \rho,$$

*then* $\|\hat{\boldsymbol{Z}}_{J^c,J}\|_{\max} < 1$ *with probability at least* $1 - 2d^{-1}$.

*Proof.* First, we can derive the upper bound of $\|\hat{\boldsymbol{Z}}_{J^c,J}\|_{\max}$ as follows:

$$\begin{aligned}
\|\hat{\boldsymbol{Z}}_{J^c,J}\|_{\max} &= \frac{1}{\rho\|\hat{\boldsymbol{x}}\|_1}\|\boldsymbol{M}_{J^c,J}\hat{\boldsymbol{x}}\hat{\boldsymbol{z}}^\top\|_{\max} = \frac{1}{\rho\|\hat{\boldsymbol{x}}\|_1} \cdot \max_{i\in J^c}\left|\sum_{j\in J} M_{i,j}\hat{x}_j\right| \\
&\leq \frac{1}{\rho\|\hat{\boldsymbol{x}}\|_1} \cdot \left(\max_{i\in J^c}\max_{j\in J}|M_{i,j}|\right) \cdot \sum_{j\in J}|\hat{x}_j| = \frac{1}{\rho} \cdot \|\boldsymbol{M}_{J^c,J}\|_{\max} \\
&= \frac{1}{\rho} \cdot \|\boldsymbol{M}_{J^c,J} - \mathbb{E}[\boldsymbol{M}_{J^c,J}] + \mathbb{E}[\boldsymbol{M}_{J^c,J}]\|_{\max} \\
&\leq \frac{1}{\rho} \cdot \|\boldsymbol{M}_{J^c,J} - \mathbb{E}[\boldsymbol{M}_{J^c,J}]\|_{\max} + \frac{1}{\rho} \cdot \|\mathbb{E}[\boldsymbol{M}_{J^c,J}]\|_{\max} \\
&\leq \frac{1}{\rho} \cdot 2\sigma\sqrt{\Delta_{\max}(\mathcal{G}_{J,J^c})\log d} + \frac{1}{\rho} \cdot \|\boldsymbol{M}^*_{J^c,J}\|_{\max}
\end{aligned}$$

where the last inequality holds with probability at least $1 - 2d^{-1}$, by Lemma 5. Hence, if the following inequality holds:

$$2\sigma\sqrt{\Delta_{\max}(\mathcal{G}_{J,J^c})\log d} + \|\boldsymbol{M}^*_{J^c,J}\|_{\max} < \rho,$$

then $\|\hat{\boldsymbol{Z}}_{J^c,J}\|_{\max} < 1$ with probability at least $1 - 2d^{-1}$.

$\square$

**Lemma 4** (Sufficient Condition for (8),(9)). *Let* $\hat{\boldsymbol{Z}}_{J^c,J^c} = \frac{1}{\rho}\left(\boldsymbol{M}_{J^c,J^c} - \mathbb{E}[\boldsymbol{M}_{J^c,J^c}]\right)$. *If the condition in Lemma 2 holds and the following inequalities hold:*

$$(1+\xi) \cdot \left(2\sigma\sqrt{\Delta_{\max}(\mathcal{G}_{J,J^c})\log d} + \|\boldsymbol{M}^*_{J^c,J}\|_2\right) \cdot (1+\sqrt{s}) \leq \frac{\phi(\mathcal{G}_{J,J})}{2s} \cdot \bar{\lambda}(\boldsymbol{M}^*_{J,J}) \cdot \left(1 - \frac{1}{\sqrt{2}}\min_{i\in J}|u_{1,i}|\right),$$

$$(1+\xi) \cdot \|\boldsymbol{M}^*_{J^c,J^c}\|_2 \leq \frac{\phi(\mathcal{G}_{J,J})}{2s} \cdot \bar{\lambda}(\boldsymbol{M}^*_{J,J}) \cdot \left(1 - \frac{1}{2\sqrt{2}}\min_{i\in J}|u_{1,i}|\right),$$

$$2\sigma\sqrt{\Delta_{\max}(\mathcal{G}_{J^c,J^c})\log d} < \rho,$$

*then* $\lambda_1(\boldsymbol{M}_{J,J} - \rho\hat{\boldsymbol{z}}\hat{\boldsymbol{z}}^\top) = \lambda_1(\boldsymbol{M} - \rho\hat{\boldsymbol{Z}})$, $\lambda_1(\boldsymbol{M}_{J,J} - \rho\hat{\boldsymbol{z}}\hat{\boldsymbol{z}}^\top) > \lambda_2(\boldsymbol{M}_{J,J} - \rho\hat{\boldsymbol{z}}\hat{\boldsymbol{z}}^\top)$ *and* $\|\boldsymbol{Z}_{J^c,J^c}\|_{\max} < 1$ *with probability at least* $1 - 2s^{-1} - 4d^{-1}$. *Here,* $\xi \geq 0$ *is a constant satisfying* $\|(\boldsymbol{A}_\mathcal{G})_{J^c,J} \circ \boldsymbol{M}^*_{J^c,J}\|_2 \leq (1+\xi) \cdot \|\boldsymbol{M}^*_{J^c,J}\|_2$ *and* $\|(\boldsymbol{A}_\mathcal{G})_{J^c,J^c} \circ \boldsymbol{M}^*_{J^c,J^c}\|_2 \leq (1+\xi) \cdot \|\boldsymbol{M}^*_{J^c,J^c}\|_2$.

*Proof.* Lemma 8 shows that if the following inequality holds:

$$\underbrace{\|\boldsymbol{M}_{J^c,J} - \rho\hat{\boldsymbol{Z}}_{J^c,J}\|_2^2}_{=:a_1} \leq \underbrace{\{\lambda_1(\boldsymbol{M}_{J,J} - \rho\hat{\boldsymbol{z}}\hat{\boldsymbol{z}}^\top) - \lambda_2(\boldsymbol{M}_{J,J} - \rho\hat{\boldsymbol{z}}\hat{\boldsymbol{z}}^\top)\}}_{=:a_2} \cdot \underbrace{\{\lambda_1(\boldsymbol{M}_{J,J} - \rho\hat{\boldsymbol{z}}\hat{\boldsymbol{z}}^\top) - \lambda_1(\boldsymbol{M}_{J^c,J^c} - \rho\hat{\boldsymbol{Z}}_{J^c,J^c})\}}_{=:a_3},$$

then $\lambda_1(\boldsymbol{M}_{J,J} - \rho\hat{\boldsymbol{z}}\hat{\boldsymbol{z}}^\top) = \lambda_1(\boldsymbol{M} - \rho\hat{\boldsymbol{Z}})$.

**Upper Bound of $a_1$:**

$$\|\boldsymbol{M}_{J^c,J} - \rho\hat{\boldsymbol{Z}}_{J^c,J}\|_2 = \left\|\boldsymbol{M}_{J^c,J} - \rho \cdot \frac{1}{\rho\|\hat{\boldsymbol{x}}\|_1}\boldsymbol{M}_{J^c,J}\hat{\boldsymbol{x}}\hat{\boldsymbol{z}}^\top\right\|_2 = \left\|\boldsymbol{M}_{J^c,J}\cdot\left(I - \frac{\hat{\boldsymbol{x}}\hat{\boldsymbol{z}}^\top}{\|\hat{\boldsymbol{x}}\|_1}\right)\right\|_2$$

$$\leq \|\boldsymbol{M}_{J^c,J}\|_2 \cdot \left\|I - \frac{\hat{\boldsymbol{x}}\hat{\boldsymbol{z}}^\top}{\|\hat{\boldsymbol{x}}\|_1}\right\|_2 \leq \|\boldsymbol{M}_{J^c,J}\|_2 \cdot \left(1 + \frac{\|\hat{\boldsymbol{x}}\|_2\|\hat{\boldsymbol{z}}\|_2}{\|\hat{\boldsymbol{x}}\|_1}\right)$$

$$\leq \|\boldsymbol{M}_{J^c,J}\|_2 \cdot (1 + \sqrt{s})$$

$$= \|\boldsymbol{M}_{J^c,J} - \mathbb{E}[\boldsymbol{M}_{J^c,J}] + \mathbb{E}[\boldsymbol{M}_{J^c,J}]\|_2 \cdot (1 + \sqrt{s})$$

$$\leq \left(\|\boldsymbol{M}_{J^c,J} - \mathbb{E}[\boldsymbol{M}_{J^c,J}]\|_2 + \|(\boldsymbol{A}_\mathcal{G})_{J^c,J} \circ \boldsymbol{M}^*_{J^c,J}\|_2\right) \cdot (1 + \sqrt{s})$$

$$\leq \left(2\sigma\sqrt{\Delta_{\max}(\mathcal{G}_{J,J^c})\log d} + (1+\xi)\cdot\|\boldsymbol{M}^*_{J^c,J}\|_2\right) \cdot (1 + \sqrt{s})$$

$$\leq (1+\xi)\cdot\left(2\sigma\sqrt{\Delta_{\max}(\mathcal{G}_{J,J^c})\log d} + \|\boldsymbol{M}^*_{J^c,J}\|_2\right) \cdot (1 + \sqrt{s})$$

where the penultimate inequality holds with probability at least $1 - 2d^{-1}$, by Lemma 5.

**Lower Bound of $a_2$:** By Weyl's inequality,

$$\lambda_1(\boldsymbol{M}_{J,J} - \rho\hat{\boldsymbol{z}}\hat{\boldsymbol{z}}^\top) - \lambda_2(\boldsymbol{M}_{J,J} - \rho\hat{\boldsymbol{z}}\hat{\boldsymbol{z}}^\top)$$

$$\geq \frac{\phi(\mathcal{G}_{J,J})}{s}\cdot\lambda_1(\boldsymbol{M}^*_{J,J}) - \frac{\phi(\mathcal{G}_{J,J})}{s}\cdot\lambda_2(\boldsymbol{M}^*_{J,J}) - 2\cdot\|\frac{\phi(\mathcal{G}_{J,J})}{s}\boldsymbol{M}^*_{J,J} - \boldsymbol{M}_{J,J} + \rho\hat{\boldsymbol{z}}\hat{\boldsymbol{z}}^\top\|_2$$

$$\geq \frac{\phi(\mathcal{G}_{J,J})}{s}\cdot\bar{\lambda}(\boldsymbol{M}^*_{J,J}) - \frac{2\phi(\mathcal{G}_{J,J})\bar{\lambda}(\boldsymbol{M}^*_{J,J})\cdot\min_{i\in J}|u_{1,i}|}{2\sqrt{2}s}$$

$$= \frac{\phi(\mathcal{G}_{J,J})}{s}\cdot\bar{\lambda}(\boldsymbol{M}^*_{J,J})\cdot\left(1 - \frac{1}{\sqrt{2}}\min_{i\in J}|u_{1,i}|\right)$$

where the second inequality holds with probability at least $1 - 2s^{-1}$, by Lemma 2.

**Lower Bound of $a_3$:** Finally, in a similar way to the above, we have that

$$\lambda_1(\boldsymbol{M}_{J,J} - \rho\hat{\boldsymbol{z}}\hat{\boldsymbol{z}}^\top) \geq \frac{\phi(\mathcal{G}_{J,J})}{s}\cdot\lambda_1(\boldsymbol{M}^*_{J,J}) - \|\frac{\phi(\mathcal{G}_{J,J})}{s}\boldsymbol{M}^*_{J,J} - \boldsymbol{M}_{J,J} + \rho\hat{\boldsymbol{z}}\hat{\boldsymbol{z}}^\top\|_2$$

$$\geq \frac{\phi(\mathcal{G}_{J,J})}{s}\cdot\bar{\lambda}(\boldsymbol{M}^*_{J,J}) - \frac{\phi(\mathcal{G}_{J,J})\bar{\lambda}(\boldsymbol{M}^*_{J,J})\cdot\min_{i\in J}|u_{1,i}|}{2\sqrt{2}s}$$

$$= \frac{\phi(\mathcal{G}_{J,J})}{s}\cdot\bar{\lambda}(\boldsymbol{M}^*_{J,J})\cdot\left(1 - \frac{1}{2\sqrt{2}}\min_{i\in J}|u_{1,i}|\right)$$

with probability at least $1 - 2s^{-1}$. Also, since $\hat{\boldsymbol{Z}}_{J^c,J^c} = \frac{1}{\rho}\left(\boldsymbol{M}_{J^c,J^c} - \mathbb{E}[\boldsymbol{M}_{J^c,J^c}]\right)$,

$$\lambda_1(\boldsymbol{M}_{J^c,J^c} - \rho\hat{\boldsymbol{Z}}_{J^c,J^c}) = \lambda_1(\mathbb{E}[\boldsymbol{M}_{J^c,J^c}]) = \lambda_1((\boldsymbol{A}_\mathcal{G})_{J^c,J^c} \circ \boldsymbol{M}^*_{J^c,J^c}) \leq (1+\xi)\cdot\|\boldsymbol{M}^*_{J^c,J^c}\|_2.$$

Hence, $a_3$ is lower-bounded by $\frac{\phi(\mathcal{G}_{J,J})}{s}\cdot\bar{\lambda}(\boldsymbol{M}^*_{J,J})\cdot\left(1 - \frac{1}{2\sqrt{2}}\min_{i\in J}|u_{1,i}|\right) - (1+\xi)\cdot\|\boldsymbol{M}^*_{J^c,J^c}\|_2$.

By using the bounds of $a_1$, $a_2$ and $a_3$, we can derive that if the following inequalities hold:

$$(1+\xi)\cdot(2\sigma\sqrt{\Delta_{\max}(\mathcal{G}_{J,J^c})\log d} + \|\boldsymbol{M}^*_{J^c,J}\|_2)\cdot(1+\sqrt{s}) \leq \frac{\phi(\mathcal{G}_{J,J})}{2s}\cdot\bar{\lambda}(\boldsymbol{M}^*_{J,J})\cdot\left(1 - \frac{1}{\sqrt{2}}\min_{i\in J}|u_{1,i}|\right),$$

$$(1+\xi)\cdot\|\boldsymbol{M}^*_{J^c,J^c}\|_2 \leq \frac{\phi(\mathcal{G}_{J,J})}{2s}\cdot\bar{\lambda}(\boldsymbol{M}^*_{J,J})\cdot\left(1 - \frac{1}{2\sqrt{2}}\min_{i\in J}|u_{1,i}|\right),$$

then $\lambda_1(\boldsymbol{M}_{J,J} - \rho\hat{\boldsymbol{z}}\hat{\boldsymbol{z}}^\top) = \lambda_1(\boldsymbol{M} - \rho\hat{\boldsymbol{Z}})$ with probability at least $1 - 2s^{-1} - 2d^{-1}$.

Lastly, by using the lower bound of $a_2$, we can derive that if $\bar{\lambda}(\boldsymbol{M}^*_{J,J}) > 0$, then $\lambda_1(\boldsymbol{M}_{J,J} - \rho\hat{\boldsymbol{z}}\hat{\boldsymbol{z}}^\top) > \lambda_2(\boldsymbol{M}_{J,J} - \rho\hat{\boldsymbol{z}}\hat{\boldsymbol{z}}^\top)$ holds with probability at least $1 - 2s^{-1}$. Note that $\bar{\lambda}(\boldsymbol{M}^*_{J,J}) > 0$ holds because $\bar{\lambda}(\boldsymbol{M}^*_{J,J}) \geq \bar{\lambda}(\boldsymbol{M}^*) > 0$ by our problem definition. Also, by using Lemma 5, we can see that if $2\sigma\sqrt{\Delta_{\max}(\mathcal{G}_{J^c,J^c})\log d} < \rho$, then $\|\boldsymbol{Z}_{J^c,J^c}\|_{\max} = \frac{1}{\rho}\|\boldsymbol{M}_{J^c,J^c} - \mathbb{E}[\boldsymbol{M}_{J^c,J^c}]\|_{\max} \leq \frac{2\sigma}{\rho}\sqrt{\Delta_{\max}(\mathcal{G}_{J^c,J^c})\log d} < 1$ holds with probability at least $1 - 2d^{-1}$. $\qquad\square$

**Step 3: Final Result**

By above lemmas, we can show the following theorem, which is the formal version of Theorem 1 in the main text.

**Theorem 4.** *Under the problem definition in Section 2, assume that the following inequalities hold:*

$$\|\boldsymbol{M}^*_{J,J}\|_2 \cdot \psi(\mathcal{G}_{J,J}) + 2\sigma\sqrt{\Delta_{\max}(\mathcal{G}_{J,J})\log s} + s\rho \leq \frac{\phi(\mathcal{G}_{J,J})\bar{\lambda}(\boldsymbol{M}^*_{J,J}) \cdot \min_{i\in J}|u_{1,i}|}{2\sqrt{2}s},$$

$$2\sigma\sqrt{\Delta_{\max}(\mathcal{G}_{J,J^c})\log d} + \|\boldsymbol{M}^*_{J^c,J}\|_{\max} < \rho,$$

$$(1+\xi)\cdot\left(2\sigma\sqrt{\Delta_{\max}(\mathcal{G}_{J,J^c})\log d} + \|\boldsymbol{M}^*_{J^c,J}\|_2\right)\cdot(1+\sqrt{s}) \leq \frac{\phi(\mathcal{G}_{J,J})}{2s}\cdot\bar{\lambda}(\boldsymbol{M}^*_{J,J})\cdot\left(1 - \frac{1}{\sqrt{2}}\min_{i\in J}|u_{1,i}|\right),$$

$$(1+\xi)\cdot\|\boldsymbol{M}^*_{J^c,J^c}\|_2 \leq \frac{\phi(\mathcal{G}_{J,J})}{2s}\cdot\bar{\lambda}(\boldsymbol{M}^*_{J,J})\cdot\left(1 - \frac{1}{2\sqrt{2}}\min_{i\in J}|u_{1,i}|\right),$$

$$2\sigma\sqrt{\Delta_{\max}(\mathcal{G}_{J^c,J^c})\log d} < \rho,$$

*where $\xi \geq 0$ is a constant satisfying $\|(\boldsymbol{A}_\mathcal{G})_{J^c,J} \circ \boldsymbol{M}^*_{J^c,J}\|_2 \leq (1+\xi)\cdot\|\boldsymbol{M}^*_{J^c,J}\|_2$ and $\|(\boldsymbol{A}_\mathcal{G})_{J^c,J^c} \circ \boldsymbol{M}^*_{J^c,J^c}\|_2 \leq (1+\xi)\cdot\|\boldsymbol{M}^*_{J^c,J^c}\|_2$. Then $\hat{\boldsymbol{X}} := \begin{pmatrix} \hat{\boldsymbol{x}}\hat{\boldsymbol{x}}^\top & 0 \\ 0 & 0 \end{pmatrix}$ with $\hat{\boldsymbol{x}}$ defined in (5) is a unique optimal solution to the problem (1), and it satisfies $\mathrm{supp}(\mathrm{diag}(\hat{\boldsymbol{X}})) = J$, with probability at least $1 - 2s^{-1} - 4d^{-1}$.*

Consider the following choice of the tuning parameter $\rho$:

$$\rho = 2\sigma\sqrt{\max\left\{\Delta_{\max}(\mathcal{G}_{J,J^c}), \Delta_{\max}(\mathcal{G}_{J^c,J^c})\right\}\log d} + \|\boldsymbol{M}^*_{J^c,J}\|_{\max}. \tag{10}$$

Then it suffices to satisfy

$$\|\boldsymbol{M}^*_{J,J}\|_2 \cdot \psi(\mathcal{G}_{J,J}) + 2\sigma\sqrt{\Delta_{\max}(\mathcal{G}_{J,J})\log s} + 2\sigma s\sqrt{\max\left\{\Delta_{\max}(\mathcal{G}_{J,J^c}), \Delta_{\max}(\mathcal{G}_{J^c,J^c})\right\}\log d} + s\|\boldsymbol{M}^*_{J^c,J}\|_{\max}$$
$$\leq \frac{\phi(\mathcal{G}_{J,J})\bar{\lambda}(\boldsymbol{M}^*_{J,J}) \cdot \min_{i\in J}|u_{1,i}|}{2\sqrt{2}s},$$

$$(1+\xi)\cdot\left(2\sigma\sqrt{\Delta_{\max}(\mathcal{G}_{J,J^c})\log d} + \|\boldsymbol{M}^*_{J^c,J}\|_2\right)\cdot(1+\sqrt{s}) \leq \frac{\phi(\mathcal{G}_{J,J})}{2s}\cdot\bar{\lambda}(\boldsymbol{M}^*_{J,J})\cdot\left(1 - \frac{1}{\sqrt{2}}\min_{i\in J}|u_{1,i}|\right),$$

$$(1+\xi)\cdot\|\boldsymbol{M}^*_{J^c,J^c}\|_2 \leq \frac{\phi(\mathcal{G}_{J,J})}{2s}\cdot\bar{\lambda}(\boldsymbol{M}^*_{J,J})\cdot\left(1 - \frac{1}{2\sqrt{2}}\min_{i\in J}|u_{1,i}|\right).$$

Note that $\min_{i\in J}|u_{1,i}| \leq \frac{1}{\sqrt{s}}$. Hence, the second and third inequalities are satisfied when

$$2\sigma\sqrt{\Delta_{\max}(\mathcal{G}_{J,J^c})\log d} + \|\boldsymbol{M}^*_{J^c,J}\|_2 \leq \frac{c_1\phi(\mathcal{G}_{J,J})\bar{\lambda}(\boldsymbol{M}^*_{J,J})\min_{i\in J}|u_{1,i}|}{s},$$

$$\frac{1}{\sqrt{s}}\cdot\|\boldsymbol{M}^*_{J^c,J^c}\|_2 \leq \frac{c_2\phi(\mathcal{G}_{J,J})\bar{\lambda}(\boldsymbol{M}^*_{J,J})\min_{i\in J}|u_{1,i}|}{s}$$

for some constants $c_1, c_2 > 0$. Therefore, the sufficient conditions hold if

$$\|\boldsymbol{M}^*_{J,J}\|_2 \cdot \psi(\mathcal{G}_{J,J}) + \sigma\sqrt{\Delta_{\max}(\mathcal{G}_{J,J})\log s} + \sigma s\sqrt{\max\left\{\Delta_{\max}(\mathcal{G}_{J,J^c}), \Delta_{\max}(\mathcal{G}_{J^c,J^c})\right\}\log d}$$
$$+ s\|\boldsymbol{M}^*_{J^c,J}\|_2 + \frac{1}{\sqrt{s}}\|\boldsymbol{M}^*_{J^c,J^c}\|_2 \leq \frac{c\phi(\mathcal{G}_{J,J})\bar{\lambda}(\boldsymbol{M}^*_{J,J}) \cdot \min_{i\in J}|u_{1,i}|}{s},$$

with some constant $c > 0$. Since $\bar{\lambda}(\boldsymbol{M}^*_{J,J}) \geq \bar{\lambda}(\boldsymbol{M}^*)$, we can replace $\bar{\lambda}(\boldsymbol{M}^*_{J,J})$ by $\bar{\lambda}(\boldsymbol{M}^*)$.

# 8 PROOF OF THEOREM 2

We make use of the following theorem to prove Theorem 2.

**Theorem 5** (Master Tail Bound for Independent Sums (Theorem 3.6 in Tropp [2012]))**.** *Consider a finite sequence $\{\boldsymbol{Z}_l\}_{l=1}^m$ of independent, random, symmetric matrices. For all $t \in \mathbb{R}$,*

$$\mathbb{P}\Big[\lambda_1\Big(\sum_{l=1}^m \boldsymbol{Z}_l\Big) \geq t\Big] \leq \inf_{\theta > 0}\Big\{e^{-\theta t} \cdot \operatorname{trexp}\Big(\sum_{l=1}^m \log \mathbb{E}e^{\theta \boldsymbol{Z}_l}\Big)\Big\}.$$

*If $\boldsymbol{Z}_l$ and $-\boldsymbol{Z}_l$ have the same distribution for all $l$, then for any $t \geq 0$,*

$$\mathbb{P}\Big[\Big\|\sum_{l=1}^m \boldsymbol{Z}_l\Big\|_2 \geq t\Big] \leq 2 \cdot \inf_{\theta > 0}\Big\{e^{-\theta t} \cdot \operatorname{trexp}\Big(\sum_{l=1}^m \log \mathbb{E}e^{\theta \boldsymbol{Z}_l}\Big)\Big\}.$$

The following theorem is a comprehensive version of Theorem 2, which includes the result of the symmetric random matrix case.

**Theorem 6** (Tail Bound for Partial Random Matrix with Independent Sub-Gaussian Entries)**.** *Consider a $m \times n$ random matrix $\boldsymbol{Z}$ whose subset of entries independently follow sub-Gaussian distributions which are symmetric about zero and have parameter $\sigma > 0$, while the other entries are zero. That is, there exists an index set $S \subseteq \{(i,j) \mid i \in [m], j \in [n]\}$ such that for $i \in [m]$ and $j \in [n]$,*

$$Z_{i,j} = \begin{cases} N_{i,j} & \text{if } (i,j) \in S \\ 0 & \text{if } (i,j) \notin S \end{cases}$$

*where each $N_{i,j}$ is symmetric about zero and satisfies $\mathbb{E}e^{\theta N_{i,j}} \leq e^{\frac{\sigma^2 \theta^2}{2}}$ for any $\theta > 0$. Then for any $t \geq 0$,*

$$\mathbb{P}[\|\boldsymbol{Z}\|_2 \geq t] \leq 2(m+n) \cdot \exp\Big(-\frac{t^2}{2\sigma^2 \Delta_{\max}(\mathcal{G}_S)}\Big),$$

*where $\mathcal{G}_S$ is a bipartite graph whose vertex and edge sets are $[m] \times [n]$ and $S$, respectively. This inequality implies that*

$$\|\boldsymbol{Z}\|_2 \leq 2\sigma\sqrt{\Delta_{\max}(\mathcal{G}_S)\log(m+n)}$$

*with probability at least $1 - 2(m+n)^{-1}$.*

*If $\boldsymbol{Z}$ is a symmetric matrix with dimension $n$, then for any $t \geq 0$,*

$$\mathbb{P}[\|\boldsymbol{Z}\|_2 \geq t] \leq 2n \cdot \exp\Big(-\frac{t^2}{2\sigma^2 \Delta_{\max}(\mathcal{G}_S)}\Big),$$

*where $\mathcal{G}_S$ is an undirected graph whose vertex and edge sets are $[n]$ and $S$, respectively. This implies that*

$$\|\boldsymbol{Z}\|_2 \leq 2\sigma\sqrt{\Delta_{\max}(\mathcal{G}_S)\log n}$$

*with probability at least $1 - 2n^{-1}$.*

*Proof.* We first consider the case that $\boldsymbol{Z}$ is a symmetric matrix with dimension $n$. We can write $\boldsymbol{Z}$ as follows:

$$\boldsymbol{Z} = \sum_{i,j \in [n]:(i,j) \in S} N_{i,j}\boldsymbol{e}_i\boldsymbol{e}_j^\top$$

$$= \sum_{\substack{i,j \in [n], i < j: \\ (i,j) \in S}} \underbrace{N_{i,j}(\boldsymbol{e}_i\boldsymbol{e}_j^\top + \boldsymbol{e}_j\boldsymbol{e}_i^\top)}_{=:\boldsymbol{W}_{i,j}} + \sum_{\substack{i \in [n]: \\ (i,j) \in S}} \underbrace{N_{i,i}\boldsymbol{e}_i\boldsymbol{e}_i^\top}_{=:\boldsymbol{W}_{i,i}},$$

which can be viewed as a sum of independent, symmetric matrices $\{\boldsymbol{W}_{i,j}\}_{i\leq j,(i,j)\in S}$. We first note that for any $\theta > 0$ and $i, j \in [n]$ such that $i < j$,

$$
\begin{aligned}
e^{\theta \boldsymbol{W}_{i,j}} &= \boldsymbol{I} + \sum_{k=1}^{\infty} \frac{(\theta \boldsymbol{W}_{i,j})^k}{k!} \\
&= \boldsymbol{I} + \sum_{k=1}^{\infty} \frac{(\theta N_{i,j})^{2k}}{(2k)!} (\boldsymbol{e}_i \boldsymbol{e}_j^\top + \boldsymbol{e}_j \boldsymbol{e}_i^\top)^{2k} + \sum_{k=1}^{\infty} \frac{(\theta N_{i,j})^{2k-1}}{(2k-1)!} (\boldsymbol{e}_i \boldsymbol{e}_j^\top + \boldsymbol{e}_j \boldsymbol{e}_i^\top)^{2k-1} \\
&= \boldsymbol{I} + \sum_{k=1}^{\infty} \frac{(\theta N_{i,j})^{2k}}{(2k)!} (\boldsymbol{e}_i \boldsymbol{e}_i^\top + \boldsymbol{e}_j \boldsymbol{e}_j^\top) + \sum_{k=1}^{\infty} \frac{(\theta N_{i,j})^{2k-1}}{(2k-1)!} (\boldsymbol{e}_i \boldsymbol{e}_j^\top + \boldsymbol{e}_j \boldsymbol{e}_i^\top) \\
&= \boldsymbol{I} + \left( \frac{e^{\theta N_{i,j}} + e^{-\theta N_{i,j}}}{2} - 1 \right) \cdot (\boldsymbol{e}_i \boldsymbol{e}_i^\top + \boldsymbol{e}_j \boldsymbol{e}_j^\top) + \left( \frac{e^{\theta N_{i,j}} - e^{-\theta N_{i,j}}}{2} \right) \cdot (\boldsymbol{e}_i \boldsymbol{e}_j^\top + \boldsymbol{e}_j \boldsymbol{e}_i^\top),
\end{aligned}
$$

and for $i \in [n]$,

$$
\begin{aligned}
e^{\theta \boldsymbol{W}_{i,i}} &= \boldsymbol{I} + \sum_{k=1}^{\infty} \frac{(\theta \boldsymbol{W}_{i,i})^k}{k!} = \boldsymbol{I} + \sum_{k=1}^{\infty} \frac{(\theta N_{i,i})^k}{k!} (\boldsymbol{e}_i \boldsymbol{e}_i^\top)^k \\
&= \boldsymbol{I} + \sum_{k=1}^{\infty} \frac{(\theta N_{i,i})^k}{k!} \boldsymbol{e}_i \boldsymbol{e}_i^\top = \boldsymbol{I} + \left( e^{\theta N_{i,i}} - 1 \right) \cdot \boldsymbol{e}_i \boldsymbol{e}_i^\top.
\end{aligned}
$$

These quantities have the expectations as follows:

$$
\begin{aligned}
\mathbb{E} e^{\theta \boldsymbol{W}_{i,j}} &= \boldsymbol{I} + (\mathbb{E} e^{\theta N_{i,j}} - 1) \cdot (\boldsymbol{e}_i \boldsymbol{e}_i^\top + \boldsymbol{e}_j \boldsymbol{e}_j^\top) \\
\mathbb{E} e^{\theta \boldsymbol{W}_{i,i}} &= \boldsymbol{I} + (\mathbb{E} e^{\theta N_{i,i}} - 1) \cdot \boldsymbol{e}_i \boldsymbol{e}_i^\top
\end{aligned}
$$

where the fact that $\mathbb{E} e^{\theta N_{i,j}} = \mathbb{E} e^{-\theta N_{i,j}}$ is used, which is because each $N_{i,j}$ is symmetric about zero. Note that each $\mathbb{E} e^{\theta \boldsymbol{W}_{i,j}}$ ($\mathbb{E} e^{\theta \boldsymbol{W}_{i,i}}$, resp.) is a diagonal matrix whose $i$-th and $j$-th ($i$-th, resp.) diagonal entries are $\mathbb{E} e^{\theta N_{i,j}}$ ($\mathbb{E} e^{\theta N_{i,i}}$, resp.) while the other diagonal entries are 1. Now we can write the summation of the logarithms of the expectations as follows:

$$
\begin{aligned}
\sum_{\substack{i,j\in[n],i<j: \\ (i,j)\in S}} \log \mathbb{E} e^{\theta \boldsymbol{W}_{i,j}} + \sum_{\substack{i\in[n]: \\ (i,j)\in S}} \log \mathbb{E} e^{\theta \boldsymbol{W}_{i,i}} &= \log \left( \prod_{\substack{i,j\in[n],i<j: \\ (i,j)\in S}} \mathbb{E} e^{\theta \boldsymbol{W}_{i,j}} \cdot \prod_{\substack{i\in[n]: \\ (i,j)\in S}} \mathbb{E} e^{\theta \boldsymbol{W}_{i,i}} \right) \\
&= \log \left( \sum_{i\in[n]} \left( \prod_{j\in[n],(i,j)\in S} \mathbb{E} e^{\theta N_{i,j}} \right) \cdot \boldsymbol{e}_i \boldsymbol{e}_i^\top \right)
\end{aligned}
$$

where the first equality holds because $\mathbb{E} e^{\theta \boldsymbol{W}_{i,j}}$'s and $\mathbb{E} e^{\theta \boldsymbol{W}_{i,i}}$'s are positive definite and commute. Hence,

$$
\begin{aligned}
\mathrm{trexp}\left( \sum_{\substack{i,j\in[n],i<j: \\ (i,j)\in S}} \log \mathbb{E} e^{\theta \boldsymbol{W}_{i,j}} + \sum_{\substack{i\in[n]: \\ (i,j)\in S}} \log \mathbb{E} e^{\theta \boldsymbol{W}_{i,i}} \right) &= \mathrm{trexp} \log \left( \sum_{i\in[n]} \left( \prod_{j\in[n],(i,j)\in S} \mathbb{E} e^{\theta N_{i,j}} \right) \cdot \boldsymbol{e}_i \boldsymbol{e}_i^\top \right) \\
&= \mathrm{tr}\left( \sum_{i\in[n]} \left( \prod_{j\in[n],(i,j)\in S} \mathbb{E} e^{\theta N_{i,j}} \right) \cdot \boldsymbol{e}_i \boldsymbol{e}_i^\top \right) = \sum_{i\in[n]} \left( \prod_{j\in[n],(i,j)\in S} \mathbb{E} e^{\theta N_{i,j}} \right).
\end{aligned}
$$

Therefore, we have that

$$
\inf_{\theta>0} \left\{ e^{-\theta t} \cdot \mathrm{trexp}\left( \sum_{\substack{i,j\in[n],i<j: \\ (i,j)\in S}} \log \mathbb{E} e^{\theta \boldsymbol{W}_{i,j}} + \sum_{\substack{i\in[n]: \\ (i,j)\in S}} \log \mathbb{E} e^{\theta \boldsymbol{W}_{i,i}} \right) \right\} = \inf_{\theta>0} \left\{ e^{-\theta t} \cdot \sum_{i\in[n]} \left( \prod_{j\in[n],(i,j)\in S} \mathbb{E} e^{\theta N_{i,j}} \right) \right\}.
$$

Since $\mathbb{E}e^{\theta N_{i,j}} \leq e^{\frac{\sigma^2\theta^2}{2}}$ for any $\theta > 0$ and $i, j \in [n]$, we can derive that

$$
\inf_{\theta>0}\left\{ e^{-\theta t} \cdot \sum_{i\in[n]}\left(\prod_{j\in[n],(i,j)\in S}\mathbb{E}e^{\theta N_{i,j}}\right)\right\} \leq \inf_{\theta>0}\left\{ e^{-\theta t}\cdot\sum_{i\in[n]}\exp\left(\frac{\sigma^2\theta^2 \#\{j\in[n]\;;\;(i,j)\in S\}}{2}\right)\right\}
$$

$$
\leq \inf_{\theta>0}\left\{ e^{-\theta t}\cdot n\cdot\exp\left(\frac{\sigma^2\theta^2\max_{i\in[n]}\#\{j\in[n]\;;\;(i,j)\in S\}}{2}\right)\right\} = \inf_{\theta>0}\left\{ n\cdot\exp\left(\frac{\sigma^2\theta^2\Delta_{\max}(\mathcal{G}_S)}{2} - \theta t\right)\right\}
$$

$$
= n\cdot\exp\left(-\frac{t^2}{2\sigma^2\Delta_{\max}(\mathcal{G}_S)}\right).
$$

Therefore, by Theorem 5,

$$
\mathbb{P}\big[\|\boldsymbol{Z}\|_2 \geq t\big] \leq 2n\cdot\exp\left(-\frac{t^2}{2\sigma^2\Delta_{\max}(\mathcal{G}_S)}\right).
$$

Next, when $\boldsymbol{Z}$ is $m\times n$ matrix, we use the fact that $\|\boldsymbol{Z}\|_2 = \left\|\begin{pmatrix}\boldsymbol{O} & \boldsymbol{Z} \\ \boldsymbol{Z}^\top & \boldsymbol{O}\end{pmatrix}\right\|_2$. We can write that

$$
\begin{pmatrix}\boldsymbol{O} & \boldsymbol{Z} \\ \boldsymbol{Z}^\top & \boldsymbol{O}\end{pmatrix} = \sum_{\substack{i\in[m],j\in[n]:\\(i,j)\in S}}\underbrace{N_{i,j}(\boldsymbol{e}_i\boldsymbol{e}_j^\top + \boldsymbol{e}_j\boldsymbol{e}_i^\top)}_{=:\boldsymbol{W}_{i,j}}
$$

which can be viewed as a sum of independent, symmetric matrices $\{\boldsymbol{W}_{i,j}\}_{i\in[m],j\in[n],(i,j)\in S}$. As we have shown before, $\mathbb{E}e^{\theta\boldsymbol{W}_{i,j}} = \boldsymbol{I} + (\mathbb{E}e^{\theta N_{i,j}} - 1)\cdot(\boldsymbol{e}_i\boldsymbol{e}_i^\top + \boldsymbol{e}_j\boldsymbol{e}_j^\top)$, and we can derive that

$$
\sum_{\substack{i\in[m],j\in[n]:\\(i,j)\in S}}\log\mathbb{E}e^{\theta\boldsymbol{W}_{i,j}} = \log\left(\prod_{\substack{i\in[m],j\in[n]:\\(i,j)\in S}}\mathbb{E}e^{\theta\boldsymbol{W}_{i,j}}\right)
$$

$$
= \log\left(\sum_{i\in[m]}\left[\prod_{j\in[n],(i,j)\in S}\mathbb{E}e^{\theta N_{i,j}}\right]\cdot\boldsymbol{e}_i\boldsymbol{e}_i^\top + \sum_{i\in[n]}\left[\prod_{j\in[m],(i,j)\in S}\mathbb{E}e^{\theta N_{i,j}}\right]\cdot\boldsymbol{e}_i\boldsymbol{e}_i^\top\right).
$$

Hence,

$$
\text{trexp}\left(\sum_{\substack{i\in[m],j\in[n]:\\(i,j)\in S}}\log\mathbb{E}e^{\theta\boldsymbol{W}_{i,j}}\right) = \text{tr}\left(\sum_{i\in[m]}\left[\prod_{j\in[n],(i,j)\in S}\mathbb{E}e^{\theta N_{i,j}}\right]\cdot\boldsymbol{e}_i\boldsymbol{e}_i^\top + \sum_{i\in[n]}\left[\prod_{j\in[m],(i,j)\in S}\mathbb{E}e^{\theta N_{i,j}}\right]\cdot\boldsymbol{e}_i\boldsymbol{e}_i^\top\right)
$$

$$
= \sum_{i\in[m]}\left[\prod_{j\in[n],(i,j)\in S}\mathbb{E}e^{\theta N_{i,j}}\right] + \sum_{i\in[n]}\left[\prod_{j\in[m],(i,j)\in S}\mathbb{E}e^{\theta N_{i,j}}\right],
$$

and we have that

$$
\inf_{\theta>0}\left\{ e^{-\theta t}\cdot\text{trexp}\left(\sum_{\substack{i\in[m],j\in[n]:\\(i,j)\in S}}\log\mathbb{E}e^{\theta\boldsymbol{W}_{i,j}}\right)\right\}
$$

$$
= \inf_{\theta>0}\left\{ e^{-\theta t}\cdot\left(\sum_{i\in[m]}\left[\prod_{j\in[n],(i,j)\in S}\mathbb{E}e^{\theta N_{i,j}}\right] + \sum_{i\in[n]}\left[\prod_{j\in[m],(i,j)\in S}\mathbb{E}e^{\theta N_{i,j}}\right]\right)\right\}.
$$

Since $\mathbb{E}e^{\theta N_{i,j}} \leq e^{\frac{\sigma^2\theta^2}{2}}$ for any $\theta > 0$ and $i,j$, we can derive that

$$\inf_{\theta>0}\left\{ e^{-\theta t} \cdot \left( \sum_{i\in[m]}\Big[\prod_{j\in[n],(i,j)\in S}\mathbb{E}e^{\theta N_{i,j}}\Big] + \sum_{i\in[n]}\Big[\prod_{j\in[m],(i,j)\in S}\mathbb{E}e^{\theta N_{i,j}}\Big]\right)\right\}$$

$$\leq \inf_{\theta>0}\left\{ e^{-\theta t} \cdot \Big[ m\cdot\exp\Big(\frac{\sigma^2\theta^2\max_{i\in[m]}\#\{j\in[n]\,;\,(i,j)\in S\}}{2}\Big)\right.$$

$$\left. + n\cdot\exp\Big(\frac{\sigma^2\theta^2\max_{i\in[n]}\#\{j\in[m]\,;\,(i,j)\in S\}}{2}\Big)\Big]\right\}$$

$$\leq \inf_{\theta>0}\left\{ e^{-\theta t}\cdot(m+n)\cdot\exp\Big(\frac{\sigma^2\theta^2\Delta_{\max}(\mathcal{G}_S)}{2}\Big)\right\} = (m+n)\cdot\exp\Big(-\frac{t^2}{2\sigma^2\Delta_{\max}(\mathcal{G}_S)}\Big).$$

Therefore, by Theorem 5,

$$\mathbb{P}\big[\|\boldsymbol{Z}\|_2 \geq t\big] \leq 2(m+n)\cdot\exp\Big(-\frac{t^2}{2\sigma^2\Delta_{\max}(\mathcal{G}_S)}\Big).$$

$\square$

**Lemma 5.** *When each $N_{i,j}$ is symmetric about zero and satisfies $\mathbb{E}e^{\theta N_{i,j}} \leq e^{\frac{\sigma^2\theta^2}{2}}$ for any $\theta > 0$,*

$$\|\mathbb{E}[\boldsymbol{M}_{J,J}] - \boldsymbol{M}_{J,J}\|_2 = \|(\boldsymbol{A}_\mathcal{G})_{J,J}\circ\boldsymbol{N}_{J,J}\|_2 \leq 2\sigma\sqrt{\Delta_{\max}(\mathcal{G}_{J,J})\log s} \qquad \text{with probability at least } 1-2s^{-1},$$

$$\|\mathbb{E}[\boldsymbol{M}_{J^c,J}] - \boldsymbol{M}_{J^c,J}\|_2 = \|(\boldsymbol{A}_\mathcal{G})_{J^c,J}\circ\boldsymbol{N}_{J^c,J}\|_2 \leq 2\sigma\sqrt{\Delta_{\max}(\mathcal{G}_{J,J^c})\log d} \qquad \text{with probability at least } 1-2d^{-1},$$

$$\|\mathbb{E}[\boldsymbol{M}_{J^c,J^c}] - \boldsymbol{M}_{J^c,J^c}\|_2 = \|(\boldsymbol{A}_\mathcal{G})_{J^c,J^c}\circ\boldsymbol{N}_{J^c,J^c}\|_2 \leq 2\sigma\sqrt{\Delta_{\max}(\mathcal{G}_{J^c,J^c})\log d} \quad \text{with probability at least } 1-2d^{-1}.$$

*Proof.* Straightforwardly, the inequalities are obtained by invoking Theorem 6. $\square$

**Lemma 6.** *Suppose that the matrix $\boldsymbol{M}^*$ is a covariance matrix, and $\boldsymbol{X}_1,\ldots,\boldsymbol{X}_n$ are drawn from the multivariate normal distribution, $N_d(\boldsymbol{0},\boldsymbol{M}^*)$. Assume that $n > d$. Consider that we observe an incomplete sample covariance matrix, $\boldsymbol{M} = \boldsymbol{A}_\mathcal{G}\circ(\frac{1}{n}\sum_{i\in[n]}\boldsymbol{X}_i\boldsymbol{X}_i^\top)$. Define $\boldsymbol{N} = \boldsymbol{M}^* - \frac{1}{n}\sum_{i\in[n]}\boldsymbol{X}_i\boldsymbol{X}_i^\top$. Then, we have*

$$\|\mathbb{E}[\boldsymbol{M}_{J,J}] - \boldsymbol{M}_{J,J}\|_2 = \|(\boldsymbol{A}_\mathcal{G})_{J,J}\circ\boldsymbol{N}_{J,J}\|_2 \leq 2\tilde{\sigma}\sqrt{\Delta_{\max}(\mathcal{G}_{J,J})\log s} \qquad \text{with probability at least } 1-O(s^{-1}),$$

$$\|\mathbb{E}[\boldsymbol{M}_{J^c,J}] - \boldsymbol{M}_{J^c,J}\|_2 = \|(\boldsymbol{A}_\mathcal{G})_{J^c,J}\circ\boldsymbol{N}_{J^c,J}\|_2 \leq 2\tilde{\sigma}\sqrt{\Delta_{\max}(\mathcal{G}_{J,J^c})\log d} \qquad \text{with probability at least } 1-O(d^{-1}),$$

$$\|\mathbb{E}[\boldsymbol{M}_{J^c,J^c}] - \boldsymbol{M}_{J^c,J^c}\|_2 = \|(\boldsymbol{A}_\mathcal{G})_{J^c,J^c}\circ\boldsymbol{N}_{J^c,J^c}\|_2 \leq 2\tilde{\sigma}\sqrt{\Delta_{\max}(\mathcal{G}_{J^c,J^c})\log d} \quad \text{with probability at least } 1-O(d^{-1}),$$

*where $\tilde{\sigma} = \|\boldsymbol{M}^*\|_2 \cdot \frac{\sqrt{d\log d}\log n}{\sqrt{n}}$.*

*Proof.* By using Theorem 1.1 in Chen et al. [2012] and Markov's inequality, we can derive that for any $t > 0$,

$$\mathbb{P}\big[\|(\boldsymbol{A}_\mathcal{G})_{J,J}\circ\boldsymbol{N}_{J,J}\|_2^2 \geq t^2\big] \leq \frac{C}{t^2}\cdot\left\{\sqrt{\frac{\|\boldsymbol{M}^*\|_{\max}}{\|\boldsymbol{M}^*\|_2}\cdot\frac{\Delta_{\max}(\mathcal{G}_{J,J})\log s}{n}} + \frac{\|\boldsymbol{M}^*\|_{\max}}{\|\boldsymbol{M}^*\|_2}\cdot\frac{\Delta_{\max}(\mathcal{G}_{J,J})\log s\log(ns)}{n}\right\}^2\cdot\|\boldsymbol{M}^*\|_2^2$$

for some positive constant $C$. Let $t = \frac{1}{2}\left\{\sqrt{\frac{\|\boldsymbol{M}^*\|_{\max}}{\|\boldsymbol{M}^*\|_2}\cdot\frac{\Delta_{\max}(\mathcal{G}_{J,J})\log s}{n}} + \frac{\|\boldsymbol{M}^*\|_{\max}}{\|\boldsymbol{M}^*\|_2}\cdot\frac{\Delta_{\max}(\mathcal{G}_{J,J})\log s\log(ns)}{n}\right\}\cdot\|\boldsymbol{M}^*\|_2\cdot\sqrt{s}.$

Then, with probability at least $1 - O(s^{-1})$,

$$\|(\boldsymbol{A}_{\mathcal{G}})_{J,J} \circ \boldsymbol{N}_{J,J}\|_2 \le \frac{1}{2}\left\{ \sqrt{\frac{\|\boldsymbol{M}^*\|_{\max}}{\|\boldsymbol{M}^*\|_2} \cdot \frac{\Delta_{\max}(\mathcal{G}_{J,J}) \log s}{n}} + \frac{\|\boldsymbol{M}^*\|_{\max}}{\|\boldsymbol{M}^*\|_2} \cdot \frac{\Delta_{\max}(\mathcal{G}_{J,J}) \log s \log(ns)}{n} \right\} \cdot \|\boldsymbol{M}^*\|_2 \cdot \sqrt{s}$$

$$= \sqrt{\Delta_{\max}(\mathcal{G}_{J,J}) \log s} \cdot \|\boldsymbol{M}^*\|_2 \cdot \frac{1}{2}\left\{ \sqrt{\frac{\|\boldsymbol{M}^*\|_{\max}}{\|\boldsymbol{M}^*\|_2} \cdot \frac{s}{n}} + \frac{\|\boldsymbol{M}^*\|_{\max}}{\|\boldsymbol{M}^*\|_2} \cdot \frac{\sqrt{s\Delta_{\max}(\mathcal{G}_{J,J}) \log s} \log(ns)}{n} \right\}$$

$$\le \sqrt{\Delta_{\max}(\mathcal{G}_{J,J}) \log s} \cdot \|\boldsymbol{M}^*\|_2 \cdot \sqrt{\frac{s}{n}} \cdot \frac{1}{2}\left\{ 1 + \sqrt{\frac{\Delta_{\max}(\mathcal{G}_{J,J})}{n}} \cdot \sqrt{\log s} \log(ns) \right\}$$

$$\le \sqrt{\Delta_{\max}(\mathcal{G}_{J,J}) \log s} \cdot \|\boldsymbol{M}^*\|_2 \cdot \sqrt{\frac{s}{n}} \cdot \frac{1}{2}\left\{ 1 + 2\sqrt{\log s} \log n \right\} \le 2\sqrt{\Delta_{\max}(\mathcal{G}_{J,J}) \log s} \cdot \|\boldsymbol{M}^*\|_2 \cdot \frac{\sqrt{s \log s} \log n}{\sqrt{n}}.$$

In a similar way, we can derive that with probability at least $1 - O(d^{-1})$,

$$\|(\boldsymbol{A}_{\mathcal{G}})_{J^c,J} \circ \boldsymbol{N}_{J^c,J}\|_2 \le 2\sqrt{\Delta_{\max}(\mathcal{G}_{J,J^c}) \log d} \cdot \|\boldsymbol{M}^*\|_2 \cdot \frac{\sqrt{d \log d} \log n}{\sqrt{n}}$$

and

$$\|(\boldsymbol{A}_{\mathcal{G}})_{J^c,J^c} \circ \boldsymbol{N}_{J^c,J^c}\|_2 \le 2\sqrt{\Delta_{\max}(\mathcal{G}_{J^c,J^c}) \log d} \cdot \|\boldsymbol{M}^*\|_2 \cdot \frac{\sqrt{d \log d} \log n}{\sqrt{n}}.$$

$\square$

## 9  PROOF OF THEOREM 3

For simplicity, let $\phi = \phi(\mathcal{G})$ and $\psi = \psi(\mathcal{G})$ in this proof. First, note that

$$\|\boldsymbol{Y} - \frac{n}{\phi} \cdot \boldsymbol{A}_{\mathcal{G}} \circ \boldsymbol{Y}\|_2 = \max_{\|\boldsymbol{y}\|_2=1} \left| \boldsymbol{y}^\top \left\{ \sum_{k \in [r]} \lambda_k(\boldsymbol{Y}) \boldsymbol{v}_k \boldsymbol{v}_k^\top - \frac{n}{\phi} \sum_{k \in [r]} \lambda_k(\boldsymbol{Y})(\boldsymbol{v}_k \boldsymbol{v}_k^\top \circ \boldsymbol{A}_{\mathcal{G}}) \right\} \boldsymbol{y} \right|$$

$$\le \max_{\|\boldsymbol{y}\|_2=1} \sum_{k \in [r]} |\lambda_k(\boldsymbol{Y})| \cdot \left| \boldsymbol{y}^\top \left\{ \boldsymbol{v}_k \boldsymbol{v}_k^\top - \frac{n}{\phi}(\boldsymbol{v}_k \boldsymbol{v}_k^\top \circ \boldsymbol{A}_{\mathcal{G}}) \right\} \boldsymbol{y} \right|$$

$$= \max_{\|\boldsymbol{y}\|_2=1} \sum_{k \in [r]} |\lambda_k(\boldsymbol{Y})| \cdot \left| (\boldsymbol{y}^\top \boldsymbol{v}_k)^2 - \frac{n}{\phi}(\boldsymbol{y} \circ \boldsymbol{v}_k)^\top \boldsymbol{A}_{\mathcal{G}}(\boldsymbol{y} \circ \boldsymbol{v}_k) \right|. \tag{11}$$

Now, we will find the upper and lower bounds of $(\boldsymbol{y}^\top \boldsymbol{v}_k)^2 - \frac{n}{\phi}(\boldsymbol{y} \circ \boldsymbol{v}_k)^\top \boldsymbol{A}_{\mathcal{G}}(\boldsymbol{y} \circ \boldsymbol{v}_k)$. Note that we can write $\boldsymbol{y} \circ \boldsymbol{v}_k = \frac{\boldsymbol{y}^\top \boldsymbol{v}_k}{n} \cdot \boldsymbol{1} + \sqrt{(\boldsymbol{y} \circ \boldsymbol{v}_k)^\top(\boldsymbol{y} \circ \boldsymbol{v}_k) - \frac{(\boldsymbol{y}^\top \boldsymbol{v}_k)^2}{n}} \cdot \boldsymbol{1}_\perp$ with $\boldsymbol{1} = (1,1,\ldots,1)^\top \in \mathbb{R}^n$ and $\boldsymbol{1}_\perp \in \mathbb{R}^n$ where $\boldsymbol{1}_\perp$ is some unit vector orthogonal to $\boldsymbol{1}$. First, we derive the lower bound of $(\boldsymbol{y}^\top \boldsymbol{v}_k)^2 - \frac{n}{\phi}(\boldsymbol{y} \circ \boldsymbol{v}_k)^\top \boldsymbol{A}_{\mathcal{G}}(\boldsymbol{y} \circ \boldsymbol{v}_k)$ as follows:

$$(\boldsymbol{y}^\top \boldsymbol{v}_k)^2 - \frac{n}{\phi}(\boldsymbol{y} \circ \boldsymbol{v}_k)^\top \boldsymbol{A}_{\mathcal{G}}(\boldsymbol{y} \circ \boldsymbol{v}_k) = (\boldsymbol{y}^\top \boldsymbol{v}_k)^2 - \frac{n}{\phi}(\boldsymbol{y} \circ \boldsymbol{v}_k)^\top (\boldsymbol{A}_{\mathcal{G}} - \boldsymbol{D}_{\mathcal{G}} + \boldsymbol{D}_{\mathcal{G}})(\boldsymbol{y} \circ \boldsymbol{v}_k)$$

$$= (\boldsymbol{y}^\top \boldsymbol{v}_k)^2 + \frac{n}{\phi}(\boldsymbol{y} \circ \boldsymbol{v}_k)^\top (\boldsymbol{D}_{\mathcal{G}} - \boldsymbol{A}_{\mathcal{G}})(\boldsymbol{y} \circ \boldsymbol{v}_k) - \frac{n}{\phi}(\boldsymbol{y} \circ \boldsymbol{v}_k)^\top \boldsymbol{D}_{\mathcal{G}}(\boldsymbol{y} \circ \boldsymbol{v}_k)$$

$$= (\boldsymbol{y}^\top \boldsymbol{v}_k)^2 + \frac{n}{\phi} \cdot \left\{ (\boldsymbol{y} \circ \boldsymbol{v}_k)^\top (\boldsymbol{y} \circ \boldsymbol{v}_k) - \frac{(\boldsymbol{y}^\top \boldsymbol{v}_k)^2}{n} \right\} \boldsymbol{1}_\perp^\top (\boldsymbol{D}_{\mathcal{G}} - \boldsymbol{A}_{\mathcal{G}}) \boldsymbol{1}_\perp - \frac{n}{\phi}(\boldsymbol{y} \circ \boldsymbol{v}_k)^\top \boldsymbol{D}_{\mathcal{G}}(\boldsymbol{y} \circ \boldsymbol{v}_k)$$

$$\ge (\boldsymbol{y}^\top \boldsymbol{v}_k)^2 + \frac{n}{\phi} \cdot \left\{ (\boldsymbol{y} \circ \boldsymbol{v}_k)^\top (\boldsymbol{y} \circ \boldsymbol{v}_k) - \frac{(\boldsymbol{y}^\top \boldsymbol{v}_k)^2}{n} \right\} \phi - \frac{n}{\phi}(\boldsymbol{y} \circ \boldsymbol{v}_k)^\top (\boldsymbol{y} \circ \boldsymbol{v}_k) \Delta_{\max}$$

$$= \frac{n(\phi - \Delta_{\max})}{\phi} \cdot (\boldsymbol{y} \circ \boldsymbol{v}_k)^\top (\boldsymbol{y} \circ \boldsymbol{v}_k) \tag{12}$$

where $\Delta_{\max} = \Delta_{\max}(\mathcal{G})$ and $\boldsymbol{D}_{\mathcal{G}}$ is a diagonal matrix whose diagonal entries are the node degrees of $\boldsymbol{A}_{\mathcal{G}}$. Similarly, we can derive the upper bound as follows:

$$
\begin{aligned}
&(\boldsymbol{y}^\top \boldsymbol{v}_k)^2 - \frac{n}{\phi}(\boldsymbol{y} \circ \boldsymbol{v}_k)^\top \boldsymbol{A}_{\mathcal{G}}(\boldsymbol{y} \circ \boldsymbol{v}_k) \\
&= (\boldsymbol{y}^\top \boldsymbol{v}_k)^2 - \frac{n}{\phi}(\boldsymbol{y} \circ \boldsymbol{v}_k)^\top (\boldsymbol{A}_{\mathcal{G}} - \boldsymbol{1}\boldsymbol{1}^\top + \boldsymbol{1}\boldsymbol{1}^\top + n\boldsymbol{I} - \boldsymbol{D}_{\mathcal{G}} - n\boldsymbol{I} + \boldsymbol{D}_{\mathcal{G}})(\boldsymbol{y} \circ \boldsymbol{v}_k) \\
&= (\boldsymbol{y}^\top \boldsymbol{v}_k)^2 - \frac{n}{\phi}(\boldsymbol{y} \circ \boldsymbol{v}_k)^\top \boldsymbol{1}\boldsymbol{1}^\top (\boldsymbol{y} \circ \boldsymbol{v}_k) - \frac{n}{\phi}(\boldsymbol{y} \circ \boldsymbol{v}_k)^\top (\boldsymbol{A}_{\mathcal{G}} - \boldsymbol{1}\boldsymbol{1}^\top + n\boldsymbol{I} - \boldsymbol{D}_{\mathcal{G}})(\boldsymbol{y} \circ \boldsymbol{v}_k) \\
&\quad + \frac{n}{\phi}(\boldsymbol{y} \circ \boldsymbol{v}_k)^\top (n\boldsymbol{I} - \boldsymbol{D}_{\mathcal{G}})(\boldsymbol{y} \circ \boldsymbol{v}_k) \\
&= \frac{\phi - n}{\phi}(\boldsymbol{y}^\top \boldsymbol{v}_k)^2 - \frac{n}{\phi} \cdot \left\{ (\boldsymbol{y} \circ \boldsymbol{v}_k)^\top (\boldsymbol{y} \circ \boldsymbol{v}_k) - \frac{(\boldsymbol{y}^\top \boldsymbol{v}_k)^2}{n} \right\} \boldsymbol{1}_\perp^\top (\boldsymbol{A}_{\mathcal{G}} - \boldsymbol{1}\boldsymbol{1}^\top + n\boldsymbol{I} - \boldsymbol{D}_{\mathcal{G}})\boldsymbol{1}_\perp \\
&\quad + \frac{n}{\phi}(\boldsymbol{y} \circ \boldsymbol{v}_k)^\top (n\boldsymbol{I} - \boldsymbol{D}_{\mathcal{G}})(\boldsymbol{y} \circ \boldsymbol{v}_k) \\
&\leq \frac{\phi - n}{\phi}(\boldsymbol{y}^\top \boldsymbol{v}_k)^2 - \frac{n}{\phi} \cdot \left\{ (\boldsymbol{y} \circ \boldsymbol{v}_k)^\top (\boldsymbol{y} \circ \boldsymbol{v}_k) - \frac{(\boldsymbol{y}^\top \boldsymbol{v}_k)^2}{n} \right\} \phi(\overline{\mathcal{G}}) + \frac{n}{\phi}(\boldsymbol{y} \circ \boldsymbol{v}_k)^\top (\boldsymbol{y} \circ \boldsymbol{v}_k)(n - \Delta_{\min}) \\
&= \frac{\phi - (n - \phi(\overline{\mathcal{G}}))}{\phi}(\boldsymbol{y}^\top \boldsymbol{v}_k)^2 + \frac{n(\Delta_{\max}(\overline{\mathcal{G}}) - \phi(\overline{\mathcal{G}}))}{\phi}(\boldsymbol{y} \circ \boldsymbol{v}_k)^\top (\boldsymbol{y} \circ \boldsymbol{v}_k) \qquad (13)
\end{aligned}
$$

where $\boldsymbol{I} \in \mathbb{R}^{n \times n}$ is an identity matrix and $\boldsymbol{1} = (1, 1, \ldots, 1)^\top \in \mathbb{R}^n$.

We can use (12) and (13) to derive the upper bound of (11). Note that

$$
\begin{aligned}
&\sum_{k \in [r]} |\lambda_k(\boldsymbol{Y})| \cdot \left| (\boldsymbol{y}^\top \boldsymbol{v}_k)^2 - \frac{n}{\phi}(\boldsymbol{y} \circ \boldsymbol{v}_k)^\top \boldsymbol{A}_{\mathcal{G}}(\boldsymbol{y} \circ \boldsymbol{v}_k) \right| \\
&\leq \sum_{k \in [r]} |\lambda_k(\boldsymbol{Y})| \cdot \max \left\{ \left| \frac{n(\phi - \Delta_{\max})}{\phi} \cdot (\boldsymbol{y} \circ \boldsymbol{v}_k)^\top (\boldsymbol{y} \circ \boldsymbol{v}_k) \right|, \left| \frac{\phi - (n - \phi(\overline{\mathcal{G}}))}{\phi}(\boldsymbol{y}^\top \boldsymbol{v}_k)^2 + \frac{n(\Delta_{\max}(\overline{\mathcal{G}}) - \phi(\overline{\mathcal{G}}))}{\phi}(\boldsymbol{y} \circ \boldsymbol{v}_k)^\top (\boldsymbol{y} \circ \boldsymbol{v}_k) \right| \right\}.
\end{aligned}
$$

First, by using (12), we have

$$
\begin{aligned}
\sum_{k \in [r]} |\lambda_k(\boldsymbol{Y})| \cdot \left| \frac{n(\phi - \Delta_{\max})}{\phi} \cdot (\boldsymbol{y} \circ \boldsymbol{v}_k)^\top (\boldsymbol{y} \circ \boldsymbol{v}_k) \right| &= \frac{n(\Delta_{\max} - \phi)}{\phi} \cdot \sum_{k \in [r]} |\lambda_k(\boldsymbol{Y})| \cdot (\boldsymbol{y} \circ \boldsymbol{v}_k)^\top (\boldsymbol{y} \circ \boldsymbol{v}_k) \\
&\leq \frac{n(\Delta_{\max} - \phi)}{\phi} \cdot \|\boldsymbol{Y}\|_2 \cdot \sum_{k \in [r]} \sum_{i \in [n]} y_i^2 v_{k,i}^2 = \frac{n(\Delta_{\max} - \phi)}{\phi} \cdot \|\boldsymbol{Y}\|_2 \cdot \sum_{i \in [n]} y_i^2 \sum_{k \in [r]} v_{k,i}^2 \\
&\leq \frac{n(\Delta_{\max} - \phi)}{\phi} \cdot \|\boldsymbol{Y}\|_2 \cdot \tau.
\end{aligned}
$$

Also, with (13), we can derive that

$$
\begin{aligned}
&\sum_{k \in [r]} |\lambda_k(\boldsymbol{Y})| \cdot \left| \frac{\phi - (n - \phi(\overline{\mathcal{G}}))}{\phi}(\boldsymbol{y}^\top \boldsymbol{v}_k)^2 + \frac{n(\Delta_{\max}(\overline{\mathcal{G}}) - \phi(\overline{\mathcal{G}}))}{\phi}(\boldsymbol{y} \circ \boldsymbol{v}_k)^\top (\boldsymbol{y} \circ \boldsymbol{v}_k) \right| \\
&= \sum_{k \in [r]} |\lambda_k(\boldsymbol{Y})| \cdot \left| (\boldsymbol{y} \circ \boldsymbol{v}_k)^\top \left\{ \frac{\phi - (n - \phi(\overline{\mathcal{G}}))}{\phi}\boldsymbol{1}\boldsymbol{1}^\top + \frac{n(\Delta_{\max}(\overline{\mathcal{G}}) - \phi(\overline{\mathcal{G}}))}{\phi}\boldsymbol{I} \right\}(\boldsymbol{y} \circ \boldsymbol{v}_k) \right| \\
&\leq \sum_{k \in [r]} |\lambda_k(\boldsymbol{Y})| \cdot (\boldsymbol{y} \circ \boldsymbol{v}_k)^\top (\boldsymbol{y} \circ \boldsymbol{v}_k) \cdot \left\| \frac{\phi - (n - \phi(\overline{\mathcal{G}}))}{\phi}\boldsymbol{1}\boldsymbol{1}^\top + \frac{n(\Delta_{\max}(\overline{\mathcal{G}}) - \phi(\overline{\mathcal{G}}))}{\phi}\boldsymbol{I} \right\|_2 \\
&\leq \|\boldsymbol{Y}\|_2 \cdot \tau \cdot \max \left\{ \left| \frac{\phi - (n - \phi(\overline{\mathcal{G}}))}{\phi} + \frac{n(\Delta_{\max}(\overline{\mathcal{G}}) - \phi(\overline{\mathcal{G}}))}{\phi} \right|, \frac{n(\Delta_{\max}(\overline{\mathcal{G}}) - \phi(\overline{\mathcal{G}}))}{\phi} \right\} \\
&= \|\boldsymbol{Y}\|_2 \cdot \tau \cdot \frac{n(\Delta_{\max}(\overline{\mathcal{G}}) - \phi(\overline{\mathcal{G}}))}{\phi}
\end{aligned}
$$

where the last equality is due to the fact that $\phi \leq n - \phi(\overline{\mathcal{G}})$ always. Therefore, we have the upper bound of $\|\boldsymbol{Y} - \frac{n}{\phi} \cdot \boldsymbol{A}_{\mathcal{G}} \circ \boldsymbol{Y}\|_2$, which is

$$
\left\| \boldsymbol{Y} - \frac{n}{\phi} \cdot \boldsymbol{A}_{\mathcal{G}} \circ \boldsymbol{Y} \right\|_2 \leq \frac{n}{\phi} \cdot \tau \|\boldsymbol{Y}\|_2 \cdot \max\{\Delta_{\max} - \phi, \ \Delta_{\max}(\overline{\mathcal{G}}) - \phi(\overline{\mathcal{G}})\} = \frac{n\tau\psi}{\phi} \cdot \|\boldsymbol{Y}\|_2.
$$

# 10   AUXILIARY LEMMAS

**Lemma 7.** *For any unit vectors $\boldsymbol{x} \in \mathbb{R}^d$ and $\boldsymbol{y} \in \mathbb{R}^d$ such that $y_i \neq 0$ for $\forall i \in [d]$, if $\|\boldsymbol{x} - \boldsymbol{y}\|_2 \leq \min_{i \in [d]} |y_i|$, then $sign(x_i) = sign(y_i)$ for $\forall i \in [d]$.*

*Proof.* If $\boldsymbol{x} = \boldsymbol{y}$, then it is trivial that $sign(x_i) = sign(y_i)$ for $\forall i \in [d]$. If $\boldsymbol{x} \neq \boldsymbol{y}$, then for any $i \in [d]$,

$$|x_i - y_i| < \|\boldsymbol{x} - \boldsymbol{y}\|_2 \leq \min_{i \in [d]} |y_i| \leq |y_i|,$$

where the first inequality is strict since both $\boldsymbol{x}$ and $\boldsymbol{y}$ are unit vectors. The above inequality implies that

$$y_i - |y_i| < x_i < y_i + |y_i|,$$

that is, $0 < x_i < 2y_i$ if $y_i > 0$, and $2y_i < x_i < 0$ if $y_i < 0$. Therefore, $sign(x_i) = sign(y_i)$ holds for any $i \in [d]$. $\qquad\square$

**Lemma 8.** *If the following inequality holds:*

$$\|\boldsymbol{M}_{J^c,J} - \rho\hat{\boldsymbol{Z}}_{J^c,J}\|_2^2 \leq \left\{\lambda_1(\boldsymbol{M}_{J,J} - \rho\hat{\boldsymbol{z}}\hat{\boldsymbol{z}}^\top) - \lambda_2(\boldsymbol{M}_{J,J} - \rho\hat{\boldsymbol{z}}\hat{\boldsymbol{z}}^\top)\right\} \cdot \left\{\lambda_1(\boldsymbol{M}_{J,J} - \rho\hat{\boldsymbol{z}}\hat{\boldsymbol{z}}^\top) - \|\boldsymbol{M}_{J^c,J^c} - \rho\hat{\boldsymbol{Z}}_{J^c,J^c}\|_2\right\},$$

*then $\lambda_1(\boldsymbol{M}_{J,J} - \rho\hat{\boldsymbol{z}}\hat{\boldsymbol{z}}^\top) = \lambda_1(\boldsymbol{M} - \rho\hat{\boldsymbol{Z}})$ where $\hat{\boldsymbol{Z}} = \begin{pmatrix} \hat{\boldsymbol{z}}\hat{\boldsymbol{z}}^\top & \hat{\boldsymbol{Z}}_{J^c,J}^\top \\ \hat{\boldsymbol{Z}}_{J^c,J} & \hat{\boldsymbol{Z}}_{J^c,J^c} \end{pmatrix}$.*

*Proof.* First, we can show that $\lambda_1(\boldsymbol{M}_{J,J} - \rho\hat{\boldsymbol{z}}\hat{\boldsymbol{z}}^\top)$ is an eigenvalue of $\boldsymbol{M} - \rho\hat{\boldsymbol{Z}}$ where its corresponding eigenvector is $(\hat{\boldsymbol{x}}^\top, \boldsymbol{0}^\top)^\top \in \mathbb{R}^d$. This is because

$$(\boldsymbol{M} - \rho\hat{\boldsymbol{Z}})\begin{pmatrix}\hat{\boldsymbol{x}} \\ 0\end{pmatrix} = \begin{pmatrix} (\boldsymbol{M}_{J,J} - \rho\hat{\boldsymbol{z}}\hat{\boldsymbol{z}}^\top)\hat{\boldsymbol{x}} \\ (\boldsymbol{M}_{J^c,J} - \rho\hat{\boldsymbol{Z}}_{J^c,J})\hat{\boldsymbol{x}} \end{pmatrix} = \lambda_1(\boldsymbol{M}_{J,J} - \rho\hat{\boldsymbol{z}}\hat{\boldsymbol{z}}^\top) \cdot \begin{pmatrix}\hat{\boldsymbol{x}} \\ 0\end{pmatrix}$$

where the last equality holds since $\hat{\boldsymbol{x}}$ is the leading eigenvector of $\boldsymbol{M}_{J,J} - \rho\hat{\boldsymbol{z}}\hat{\boldsymbol{z}}^\top$ and

$$(\boldsymbol{M}_{J^c,J} - \rho\hat{\boldsymbol{Z}}_{J^c,J})\hat{\boldsymbol{x}} = \boldsymbol{M}_{J^c,J}\hat{\boldsymbol{x}} - \rho \cdot \frac{1}{\rho\|\hat{\boldsymbol{x}}\|_1}\boldsymbol{M}_{J^c,J}\hat{\boldsymbol{x}} \cdot \|\hat{\boldsymbol{x}}\|_1 = 0.$$

Now, it is sufficient to show that for all $\boldsymbol{y} = (\boldsymbol{y}_1^\top, \boldsymbol{y}_2^\top)^\top$ such that $\boldsymbol{y}_1 \in \mathbb{R}^s$, $\boldsymbol{y}_2 \in \mathbb{R}^{d-s}$, $\|\boldsymbol{y}_1\|_2^2 + \|\boldsymbol{y}_2\|_2^2 = 1$ and $\hat{\boldsymbol{x}}^\top\boldsymbol{y}_1 = 0$,

$$\boldsymbol{y}^\top(\boldsymbol{M} - \rho\hat{\boldsymbol{Z}})\boldsymbol{y} \leq \lambda_1(\boldsymbol{M}_{J,J} - \rho\hat{\boldsymbol{z}}\hat{\boldsymbol{z}}^\top),$$

which implies that $\lambda_1(\boldsymbol{M}_{J,J} - \rho\hat{\boldsymbol{z}}\hat{\boldsymbol{z}}^\top)$ is the largest eigenvalue of $\boldsymbol{M} - \rho\hat{\boldsymbol{Z}}$. Note that

$$\boldsymbol{y}^\top(\boldsymbol{M} - \rho\hat{\boldsymbol{Z}})\boldsymbol{y} = \boldsymbol{y}_1^\top(\boldsymbol{M}_{J,J} - \rho\hat{\boldsymbol{z}}\hat{\boldsymbol{z}}^\top)\boldsymbol{y}_1 + 2\boldsymbol{y}_2^\top(\boldsymbol{M}_{J^c,J} - \rho\hat{\boldsymbol{Z}}_{J^c,J})\boldsymbol{y}_1 + \boldsymbol{y}_2^\top(\boldsymbol{M}_{J^c,J^c} - \rho\hat{\boldsymbol{Z}}_{J^c,J^c})\boldsymbol{y}_2$$

$$\leq \lambda_2(\boldsymbol{M}_{J,J} - \rho\hat{\boldsymbol{z}}\hat{\boldsymbol{z}}^\top) \cdot \|\boldsymbol{y}_1\|_2^2 + 2\|\boldsymbol{M}_{J^c,J} - \rho\hat{\boldsymbol{Z}}_{J^c,J}\|_2 \cdot \|\boldsymbol{y}_1\|_2 \cdot \|\boldsymbol{y}_2\|_2 + \lambda_1(\boldsymbol{M}_{J^c,J^c} - \rho\hat{\boldsymbol{Z}}_{J^c,J^c}) \cdot \|\boldsymbol{y}_2\|_2^2$$

$$= \lambda_2(\boldsymbol{M}_{J,J} - \rho\hat{\boldsymbol{z}}\hat{\boldsymbol{z}}^\top) \cdot (1 - \|\boldsymbol{y}_2\|_2^2) + 2\|\boldsymbol{M}_{J^c,J} - \rho\hat{\boldsymbol{Z}}_{J^c,J}\|_2 \cdot \sqrt{1 - \|\boldsymbol{y}_2\|_2^2} \cdot \|\boldsymbol{y}_2\|_2 + \lambda_1(\boldsymbol{M}_{J^c,J^c} - \rho\hat{\boldsymbol{Z}}_{J^c,J^c}) \cdot \|\boldsymbol{y}_2\|_2^2$$

$$= \lambda_2(\boldsymbol{M}_{J,J} - \rho\hat{\boldsymbol{z}}\hat{\boldsymbol{z}}^\top) + (\lambda_1(\boldsymbol{M}_{J^c,J^c} - \rho\hat{\boldsymbol{Z}}_{J^c,J^c}) - \lambda_2(\boldsymbol{M}_{J,J} - \rho\hat{\boldsymbol{z}}\hat{\boldsymbol{z}}^\top)) \cdot \|\boldsymbol{y}_2\|_2^2$$

$$\quad + 2\|\boldsymbol{M}_{J^c,J} - \rho\hat{\boldsymbol{Z}}_{J^c,J}\|_2 \cdot \sqrt{\|\boldsymbol{y}_2\|_2^2 \cdot (1 - \|\boldsymbol{y}_2\|_2^2)}$$

$$= \lambda_2(\boldsymbol{M}_{J,J} - \rho\hat{\boldsymbol{z}}\hat{\boldsymbol{z}}^\top) + (\lambda_1(\boldsymbol{M}_{J^c,J^c} - \rho\hat{\boldsymbol{Z}}_{J^c,J^c}) - \lambda_2(\boldsymbol{M}_{J,J} - \rho\hat{\boldsymbol{z}}\hat{\boldsymbol{z}}^\top)) \cdot t + 2\|\boldsymbol{M}_{J^c,J} - \rho\hat{\boldsymbol{Z}}_{J^c,J}\|_2 \cdot \sqrt{t \cdot (1 - t)}$$

where $0 \leq t := \|\boldsymbol{y}_2\|_2^2 \leq 1$. The first inequality holds since $\boldsymbol{y}_1/\|\boldsymbol{y}_1\|_2$ is orthogonal to $\hat{\boldsymbol{x}}$, the leading eigenvector of $\boldsymbol{M}_{J,J} - \rho\hat{\boldsymbol{z}}\hat{\boldsymbol{z}}^\top$. The above upper bound of $\boldsymbol{y}^\top(\boldsymbol{M} - \rho\hat{\boldsymbol{Z}})\boldsymbol{y}$ implies that if the following inequality holds for any $t \in [0, 1]$:

$$\lambda_2(\boldsymbol{M}_{J,J} - \rho\hat{\boldsymbol{z}}\hat{\boldsymbol{z}}^\top) + (\lambda_1(\boldsymbol{M}_{J^c,J^c} - \rho\hat{\boldsymbol{Z}}_{J^c,J^c}) - \lambda_2(\boldsymbol{M}_{J,J} - \rho\hat{\boldsymbol{z}}\hat{\boldsymbol{z}}^\top)) \cdot t + 2\|\boldsymbol{M}_{J^c,J} - \rho\hat{\boldsymbol{Z}}_{J^c,J}\|_2 \cdot \sqrt{t \cdot (1 - t)}$$
$$\leq \lambda_1(\boldsymbol{M}_{J,J} - \rho\hat{\boldsymbol{z}}\hat{\boldsymbol{z}}^\top),$$

then $\lambda_1(\boldsymbol{M}_{J,J} - \rho\hat{\boldsymbol{z}}\hat{\boldsymbol{z}}^\top)$ is the largest eigenvalue of $\boldsymbol{M} - \rho\hat{\boldsymbol{Z}}$. From Lemma 9, we have that if the following inequality holds:

$$\|\boldsymbol{M}_{J^c,J} - \rho\hat{\boldsymbol{Z}}_{J^c,J}\|_2^2 \leq \left\{\lambda_1(\boldsymbol{M}_{J,J} - \rho\hat{\boldsymbol{z}}\hat{\boldsymbol{z}}^\top) - \lambda_2(\boldsymbol{M}_{J,J} - \rho\hat{\boldsymbol{z}}\hat{\boldsymbol{z}}^\top)\right\} \cdot \left\{\lambda_1(\boldsymbol{M}_{J,J} - \rho\hat{\boldsymbol{z}}\hat{\boldsymbol{z}}^\top) - \lambda_1(\boldsymbol{M}_{J^c,J^c} - \rho\hat{\boldsymbol{Z}}_{J^c,J^c})\right\},$$

then $\lambda_1(\boldsymbol{M}_{J,J} - \rho\hat{\boldsymbol{z}}\hat{\boldsymbol{z}}^\top) = \lambda_1(\boldsymbol{M} - \rho\hat{\boldsymbol{Z}})$. $\qquad\square$

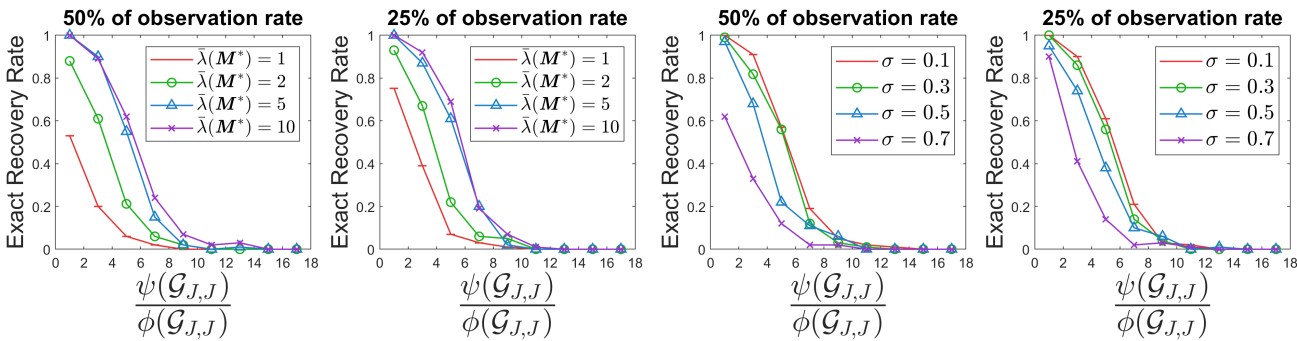

Figure 6: Rate of exact recovery of $J$ versus $\frac{\psi(\mathcal{G}_{J,J})}{\phi(\mathcal{G}_{J,J})}$. Two plots on the left present results for different $\bar{\lambda}(\boldsymbol{M}^*)$ where $\sigma = 0$. Two plots on the right present results for different $\sigma$ where $\bar{\lambda}(\boldsymbol{M}^*) = 20$.

**Lemma 9.** *Assume* $a \neq 0$. *If* $a^2 \leq c(b + c)$ *holds, then* $2a\sqrt{t(1-t)} \leq bt + c$ *for all* $t \in [0,1]$.

*Proof.*

$$
2a\sqrt{t(1-t)} \leq bt + c \quad \text{for all } t \in [0,1]
$$
$$
\Leftarrow 4a^2 t(1-t) \leq (bt+c)^2, \ bt + c \geq 0 \quad \text{for all } t \in [0,1]
$$
$$
\Leftrightarrow (4a^2 + b^2)\left(t - \frac{2a^2 - bc}{4a^2 + b^2}\right)^2 + c^2 - \frac{(2a^2 - bc)^2}{4a^2 + b^2} \geq 0, \ bt + c \geq 0 \quad \text{for all } t \in [0,1]
$$
$$
\Leftarrow c^2 - \frac{(2a^2 - bc)^2}{4a^2 + b^2} \geq 0, \ c \geq 0, \ b + c \geq 0
$$
$$
\Leftrightarrow a^2 \leq c(b + c).
$$

$\square$

## 11 ADDITIONAL SIMULATION RESULTS

Here, we present simulation results for cases where the density of the observation graph varies. Except for the density of the observation graph, we use the same setting as in Section 5.1.

First, we compare two cases where the number of observed entries is either 625 or 1250 (i.e., the observation rate is 25% or 50%.) Figure 6 shows the results where we fix the noise parameter $\sigma$ as 0 (noiseless) and try different spectral gaps $\bar{\lambda}(\boldsymbol{M}^*) \in \{1, 2, 5, 10\}$, and the results where we fix the spectral gap $\bar{\lambda}(\boldsymbol{M}^*)$ as 20 and try different noise parameters $\sigma \in \{0.1, 0.3, 0.5, 0.7\}$. We can see that when the values of $\frac{\psi(\mathcal{G}_{J,J})}{\phi(\mathcal{G}_{J,J})}$ are similar, the performance is worse in the case that more entries are observed. This shows that observing more entries outside the relevant sub-matrix negatively affects the performance of the algorithm.

However, when the observation graph is dense enough on the relevant sub-matrix and there is no noise, observing the entries completely outside the relevant sub-matrix is helpful. The first and second plots in Figure 7 show the results where we fix the noise parameter $\sigma$ as 0 (noiseless) and try different spectral gaps $\bar{\lambda}(\boldsymbol{M}^*) \in \{1, 2, 5, 10\}$. The first plot is of the case that the overall observation rate is 50%, and the second plot is of the case that the observation rate on the relevant sub-matrix is 50% and the entries outside the relevant sub-matrix are completely observed. We can check that when $\frac{\psi(\mathcal{G}_{J,J})}{\phi(\mathcal{G}_{J,J})}$ is sufficiently small, the performance is better in the second case.

On the other hand, when there is noise in the data, the opposite observation happens. The third plot in Figure 7 presents the results where $\sigma$ is set to be 0.1. We can see that the performance is worse in this case unless the spectral gap is sufficiently large and $\frac{\psi(\mathcal{G}_{J,J})}{\phi(\mathcal{G}_{J,J})}$ is sufficiently small. That is, when data is noisy, observing the entries outside the relevant sub-matrix can negatively affect the performance of the algorithm.

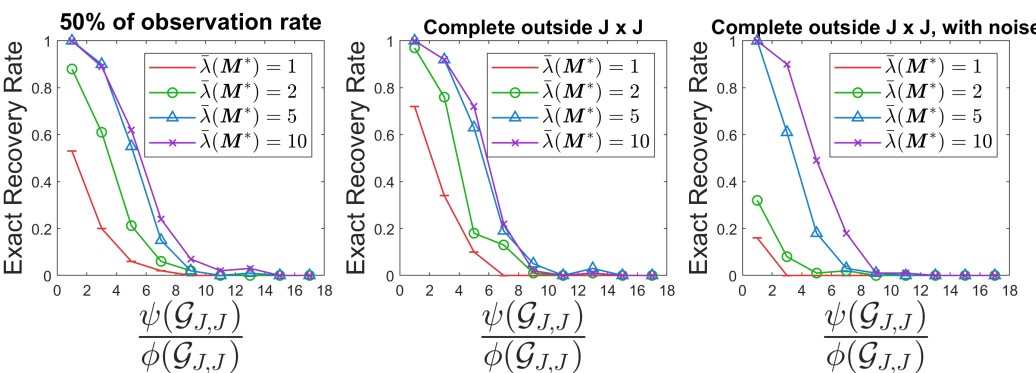

Figure 7: Rate of exact recovery of $J$ versus $\frac{\psi(\mathcal{G}_{J,J})}{\phi(\mathcal{G}_{J,J})}$. First two plots present results for different $\bar{\lambda}(\boldsymbol{M}^*)$ where $\sigma = 0$. Last plot presents results for different $\bar{\lambda}(\boldsymbol{M}^*)$ where $\sigma = 0.1$.