# OpenReview forum: "Support Recovery in Sparse PCA with General Missing Data"
_auai.org/UAI/2024/Conference — UAI 2024 oral_

### Official Review · Reviewer_xsvH · 2024-03-15

**Q2-1 Originality-Novelty:** 3
**Q2-2 Correctness-Technical Quality:** 4
**Q2-5 Clarity Of Writing:** 4

**Q1 Summary And Contributions:**

The authors study the exact recovery of the support of the sparse leading eigenvector of a matrix M using a semi-definite relaxation (SDP) relaxation of the sparse principal component analysis (s-PCA). Only an incomplete and noisy observation of M is observed. Numerical simulations are provided.

**Q2-3 Extent To Which Claims Are Supported By Evidence:**

4: Excellent: all claims are supported by very convincing evidence (in the form of comprehensive experimental evaluation, rigorous mathematical proofs, detailed (pseudo-)code, precise references, well-motivated and realistic assumptions) and the authors deliver what they promise.

**Q2-4 Reproducibility:**

2: Fair: key resources (e.g. proofs, code, data) are unavailable but key details (e.g. proof sketches, experimental setup) are sufficiently well-described for an expert to confidently reproduce the main results.

**Q3 Main Strengths:**

While dense, the paper is well-written and the setting of noise and incomplete entries in an arbitrary position is an important addition to the sparse PCA literature.
The experiments are convincing, and the comparison with other sparse PCA shows that SDP tends to better recover the support.

**Q4 Main Weakness:**

Theorem 2 (and its slight generalisation in the Appendix as Theorem 6) is not new and can be improved.

Firstly, it appears that Theorem 2 is a restatement of Eq. (4.6) in Application 4.3 in [Tropp 2012].

Then I believe that, by using Corollary 3.9 in [Bandeira 2019], one could show that || Z || \le C ( \sqrt{ \Delta_{\max} } + \sqrt{\log n} ) with probability at least 1 - 1/n, for some C > 0. This improves the \sqrt{ \Delta_{\max} * \log n }.

[Tropp 2012] Tropp, J. A. (2012). User-friendly tail bounds for sums of random matrices. Foundations of computational mathematics, 12, 389-434.

[Bandeira 2019] Afonso S. Bandeira. Ramon van Handel. "Sharp nonasymptotic bounds on the norm of random matrices with independent entries." Ann. Probab. 44 (4) 2479 - 2506, July 2016. https://doi.org/10.1214/15-AOP1025

**Q5 Detailed Comments To The Authors:**

Additional comments:

* The authors focus on the recovery of the support. Some papers also study the recovery of the signed support of the maximal eigenvector  (e.g., [Amini 2009]). Can something be said on the sign of the recovered eigenvector here? Remark 3 mentions a bound on the ell-2 error.

* Could one recover more than one eigenvector?

Minor typos:

Intro: '...the assumptions could be lack of ...' --> extra 'be'

Section 3: 'the SDP is computationally friendly' --> Can one be more specific here? (in terms of complexity, or empirical running size). Would SDP scale on large data sets (e.g. d > 100,000)?


[Amini 2009]  Arash A. Amini. Martin J. Wainwright. "High-dimensional analysis of semidefinite relaxations for sparse principal components." Ann. Statist. 37 (5B) 2877 - 2921, October 2009. https://doi.org/10.1214/08-AOS664

**Q9 Complying With Reviewing Instructions:**

Yes

---

> ### Author Rebuttal · Authors · 2024-04-09
>
> We thank the reviewer for the thoughtful feedback.
>
> >Theorem 2 (and its slight generalisation in the Appendix as Theorem 6) is not new and can be improved.
>
> We appreciate it. We will include the papers the reviewer mentioned in the references.
>
> >Can something be said on the sign of the recovered eigenvector here? Remark 3 mentions a bound on the $\ell_2$ error.
>
> Yes, we can.
> Under the sufficient condition,
> the optimal solution $\hat{\pmb{X}}$ can be expressed as
> $\hat{\pmb{X}} = ( \hat{{x}} \hat{{x}}^\top, 0 ; 0, 0 )$,
> where $\hat{{x}}$ satisfies
> sign$(\hat{{x}}; {0})$ = sign$(u_{1})$ or -sign$(u_{1})$.
> (Please check Lemma 2 in the supplementary material for detail.)
>
>
> >Could one recover more than one eigenvector?
>
> We have provided a remark regarding cases where multiple leading eigenvectors are sparse on the first page of the supplementary material. For more details, please see Section 7 of the supplementary material.
>
>
> >Section 3: 'the SDP is computationally friendly' Can one be more specific here? (in terms of complexity or empirical running size). Would SDP scale on large data sets (e.g., d $>$ 100,000)?
>
> The argument that the SDP algorithm is computationally friendly is based on a relatively recent development of the SDP solver studied by Yurtsever et al. (2021).
> They have introduced a solver called SketchyCGAL, which is applicable to all SDP problems satisfying strong duality.
> Theoretical and experimental results demonstrate that this solver can solve SDP problems much faster than existing solvers and requires far less storage.
> Specifically, they experimentally showed that the algorithm can handle SDP instances on a laptop where the matrix has over $10^{14}$ entries.
> Our experiments did not involve handling large datasets, and since the efficiency of the method is not the main focus of our study, we did not test this solver.
> For further details, please refer to Yurtsever et al. (2021).

---

### Official Review · Reviewer_v9AJ · 2024-03-15

**Q2-1 Originality-Novelty:** 3
**Q2-2 Correctness-Technical Quality:** 3
**Q2-5 Clarity Of Writing:** 2

**Q1 Summary And Contributions:**

This paper analyses the standard SDP formulation of the sparse PCA problem when entries are missing, replacing missing entries by zeros. Although the paper proposes a nice theoretical analysis in the case of missing data, I have quite a few comments/questions which makes me doubt whether the paper can be accepted in its present form. In particular, the assumptions are strong and somewhat counter-intuitive and unlikely to be satisfies; in particular:
- there exists a sparse eigenvector x.
- the observed entries should be well balanced in the submatrix M(I,I) where I is the support of x.

**Q2-3 Extent To Which Claims Are Supported By Evidence:**

3: Good: the main claims are supported by convincing evidence (in the form of adequate experimental evaluation, proofs, (pseudo-)code, references, assumptions).

**Q2-4 Reproducibility:**

3: Good: key resources (e.g. proofs, code, data) are available and key details (e.g. proofs, experimental setup) are sufficiently well-described for competent researchers to confidently reproduce the main results.

**Q3 Main Strengths:**

The authors prove the recovery of the SDP algorithm to recover the sparse eigenvector of a matrix with missing entries.

**Q4 Main Weakness:**

The assumptions are, in my opinion, not all reasonable. Also, a more natural model should be considered when entries are missing, instead of simply replacing the missing entries by zero. See below for more details.

**Q5 Detailed Comments To The Authors:**

Major comments:
- A central assumption in the paper is that the first eigenvector of the matrix $M^*$ is sparse. This is rather unlikely to happen in practice. Let $u$ be an eigenvector with support $I$, and denote $I_b$ the complement of $I$. We must have $M^*(I_b,I) u(I) = \lambda u(I_b) = 0$. This means that $u(I)$ must be orthogonal to all rows of $M^*(I_b,I)$. If $|I|$ is small hence $|I_b|$ is large, I doubt this is likely to happen in practice. In any case, it would be good to discuss this assumption in details. In fact, in the previous works I am aware of, this assumption is not made: authors look for the best sparse vectors (meaning one that maximizes $x^T M^* x$) but they do not assume the unconstrained solution is sparse.
- Authors replace missing entries in M by zeros. This makes sense if the average entries of M are zero, which I suppose is reasonable in some applications. However, in my opinion, it would make more sense to consider a model of the type min_{x, \lambda, ||x||_2=1} \sum_{(i,j) in \Omega} (M-\lambda*xx^T)_{ij}^2, leading to the following  equivalent formulation (after some calculations): max_{x, ||x||_2 = 1} (x^T M x)^2 / (sum_{(i,j) in \Omega} x_i^2 x_j^2), where missing entries in M are replaced by zeros (so that x^T M x = sum_{(i,j) in Omega} x_i M(i,j) x_j).
- Authors need to assume that the nodes in $M(I,I)$ have well connected and have similar degrees. This is because missing entries are not properly taken into account simply being replaced by zeros (cf. my comment above). In fact, it would make much more sense that "the more observations, the better" which is not the case here. If more entries are observed, this could be detrimental if they are not well balanced between nodes. In other words, if $M(I,I)$ satisfies the assumptions but we end up having additional observed entries, this could be detrimental to the algorithm/theorem, this is rather undesirable, in my opinion. In fact, one could obtain better results by removing observed entries in M(I,I) to make it more balanced, this is very odd to me.
- In the numerical experiments, authors compare several approaches, including a two-step approach: first impute, then compute the sparse eigenvector. For imputation, authors use a rather naive approach, based on the minimization of the nuclear norm under an exact recovery of all observed entries. This is not what is done in practice, especially in the presence of noise. There are many more sophisticated and more effective methods to impute missing entries in a matrix (this is a field of research on its own), and using such a naive approach is not reasonable in my opinion.


Minor comments:
- The paper is not well written, many sentence should be clarified/rewritten, e.g.,
* "Moreover, even though verification were feasible, the theorems under the assumptions could be lack of universality"
* "result yielded from matrix completion"
* "under general missingness" What does this mean?
* What is the "primal-dual witness framework" exactly?
* "These two results can be widely used in other matrix-related problems with general missing data." How can they be used? What is general missing data?
* "where r is rank of M"
- "SDP algorithm is computationally friendly": It is a common issue to solve large-scale SDPs. Authors should comment on that.
- Can you give a reference for Algebraic Connectivity?
- Top of right column on page 4: authors use both A and A_G, this is confusing.
- The choice of the tuning parameter rho is not very clear... What is C_rho exactly?
- I think there is a typo here: $\rho \in$ {0.025, 0.5, ..., 1}. What values are used? This is unclear.
- The entry-wise noise is random N(0,0.1). It is difficult to quantify the SNR with this. It would be useful to mention the (expected) norm of the noise vs. the norm of the noiseless matrix.

**Q9 Complying With Reviewing Instructions:**

Yes

---

> ### Author Rebuttal · Authors · 2024-04-09
>
> We thank the reviewer for the thoughtful feedback.
>
> * Contrary to the reviewer's understanding, many prior studies on sparse PCA have assumed strict sparsity of the first principal component [1-7]. Without this assumption, we can only discuss approximations to the principal component, which is weaker than the exact support recovery we've established.
> Relaxing the strict sparsity assumption could lead to weaker approximation guarantees in our work, but it's not the primary focus of the paper and may be explored in future work.
> Furthermore, in our semi-synthetic data analysis, we empirically showed that our approach performs effectively even without strict sparsity.
> In the experiment, the true leading eigenvector of the complete covariance matrix is not exactly sparse.
> We aimed to recover six entries with the largest magnitudes.
> * The model proposed by the reviewer is only for the original PCA problem.
> By incorporating an $\ell_1$ penalty into the suggestion to address the sparsity, we can formulate the optimization problem as follows:
> max_{x, ||x||_2 = 1} \sum_{(i,j) \in \Omega} x_i M_{i,j} x_j - \lambda \cdot ||x||_1^2,
> which is equivalent to
> max_{x, ||x||_2 = 1} < P_\Omega(M), x x^T > - \lambda \cdot ||x||_1^2.
> Here, P_\Omega(M) refers to the same matrix as $M$, but with missing entries replaced by zeros.
> This optimization problem aligns precisely with the consideration in our paper, where we employed a semidefinite relaxation for this.
> That is, our approach and the reviewer's suggestion are not very different.
> We note that replacing missing entries with zeros means disregarding them during the optimization process, under a linear objective function.
> * Our theorem suggests that the entries of $M_{J,J}$ need to be not only observed evenly but also abundantly.
> The algebraic connectivity in our theorem is linked to the observation rate;
> as more entries are observed, the algebraic connectivity tends to increase.
> Therefore, our theorem implies that removing observed entries in $M_{J,J}$ could potentially harm the algorithm, which is not unexpected.
> * Even sophisticated matrix completion methods are unlikely to outperform our approach, as they typically assume a low-rank structure in the target matrix, which does not apply to our dataset.
> We've tested a recent matrix completion algorithm [8], and the results are shown below.
> The first row shows the existing results in Figure 5 of our paper, indicating the exact recovery rates of the SDP with nuclear norm minimization,
> while the second row presents the outcomes of applying the algorithm of [8].
> Despite the use of an alternative approach, the results did not show improvement.
>
> |$\frac{\psi(G_{J,J})}{\phi(G_{J,J})}$|0.1|0.3|0.5|0.7|0.9|1.1|1.3|1.5|1.7|1.9|2.1|
> |-|-|-|-|-|-|-|-|-|-|-|-|
> |Original | 0.41 | 0.30  | 0.35 | 0.43 | 0.33 | 0.29 | 0.27 | 0.35 | 0.32 | 0.34 | 0.23 |
> |Alternative | 0.49 | 0.38 | 0.37 | 0.25 | 0.21 | 0.31 | 0.22 | 0.19 | 0.11 | 0.15 | 0.14 |
>
> * 'under general missingness' means that we do not consider any specific model assumption on the data missing scheme.
> * The primal-dual witness technique is a commonly used method for proving the consistency of variable selection in sparse high-dimensional estimation problems. Please refer to Wainwright (2009) for more details.
> * Theorems 2 and 3 can be applied to problems involving incomplete matrices, such as matrix recovery.
> * Regarding the computational efficiency of the SDP algorithm, please check the final paragraph of our response to the Reviewer xsvH.
> * Please refer to the survey paper by [9] for Algebraic Connectivity.
> * On page 4, $A$ should have been $A_G$. We'll correct this.
> * $C_\rho$ is an AIC-type criterion which balances the quality of the estimate and the complexity of the solution.
> * $\rho \in$ {0.025, 0.5, ..., 1} is not a typo. We tried all $\rho$ values from $0.025$ to $1$, then choose the best $\rho$ based on $C_\rho$ criterion. That is, the choice of $\rho$ is data-dependent.
> * In Section 5.2, the ratio of the norm of the noise to the norm of the noiseless matrix is about $0.153$. We'll add this information.
>
> [1] Amini \& Wainwright (2008). High-dimensional analysis of semidefinite relaxations for sparse principal components.
>
> [2] Vu et al. (2013). Fantope projection and selection: A near-optimal convex relaxation of sparse PCA.
>
> [3] Gu et al. (2014). Sparse PCA with oracle property.
>
> [4] Wang et al. (2014). Tighten after relax: Minimax-optimal sparse PCA in polynomial time.
>
> [5] Deshp \& Montanari (2016). Sparse PCA via covariance thresholding.
>
> [6] Gataric et al. (2020). Sparse principal component analysis via axis-aligned random projections.
>
> [7] Agterberg \& Sulam (2022). Entrywise recovery guarantees for sparse pca via sparsistent algorithms.
>
> [8] Tong et al. (2021). Accelerating ill-conditioned low-rank matrix estimation via scaled gradient descent.
>
> [9] De Abreu (2007). Old and new results on algebraic connectivity of graphs.

---

### Official Review · Reviewer_J2E4 · 2024-03-22

**Q2-1 Originality-Novelty:** 3
**Q2-2 Correctness-Technical Quality:** 3
**Q2-5 Clarity Of Writing:** 3

**Q1 Summary And Contributions:**

This paper examines the support recovery problem in sparse PCA with incomplete and noisy data, under a general sampling scheme. The authors consider a practical algorithm based on a semidefinite relaxation of the ℓ1-regularized PCA problem and provide sufficient conditions where the algorithm can exactly recover the true support with high probability. The authors demonstrate that the algorithm works well. Empirical justification is provided for the theorem through synthetic data analysis, showing that the SDP algorithm outperforms several other sparse PCA approaches, especially when the observation graph has favorable structural properties.

**Q2-3 Extent To Which Claims Are Supported By Evidence:**

3: Good: the main claims are supported by convincing evidence (in the form of adequate experimental evaluation, proofs, (pseudo-)code, references, assumptions).

**Q2-4 Reproducibility:**

3: Good: key resources (e.g. proofs, code, data) are available and key details (e.g. proofs, experimental setup) are sufficiently well-described for competent researchers to confidently reproduce the main results.

**Q3 Main Strengths:**

The main strength of the paper lies in its rigorous examination and solution of the support recovery problem within sparse PCA under challenging conditions of incomplete and noisy data. The authors propose a practical algorithm based on semidefinite relaxation, offering a novel approach to address this issue.

**Q4 Main Weakness:**

The topic of sparse PCA has already a fairly large literature.
The paper focuses on the case in which the first eigenvector is sparse.

**Q5 Detailed Comments To The Authors:**

It would be great if the authors could provide more detail on how to handle spase PCA when the number of sparse leading eigenvectors is larger than one.

**Q9 Complying With Reviewing Instructions:**

Yes

---

> ### Author Rebuttal · Authors · 2024-04-09
>
> We thank the reviewer for the thoughtful feedback.
>
> >It would be great if the authors could provide more detail on how to handle sparse PCA when the number of sparse leading eigenvectors is larger than one.
>
> We have provided a remark regarding cases where multiple leading eigenvectors are sparse on the first page of the supplementary material. For more details, please see Section 7 of the supplementary material.

---

### Official Review · Reviewer_4AgQ · 2024-03-22

**Q2-1 Originality-Novelty:** 3
**Q2-2 Correctness-Technical Quality:** 3
**Q2-5 Clarity Of Writing:** 4

**Q1 Summary And Contributions:**

The paper studies the problem of support recovery of the first principal component in sparse principal component analysis under missingess. Novelty resides in that no missingness mechanism is assumed. Authors propose a characterization of the missingness pattern through a graph and show that under conditions on its algebraic connectivity, its node degree distribution (irregularity) and spectral gap, support recovery is achieved with high probability by solving a semidefinite relaxation of the L1-regularised PCA problem (SDP algorithm). Experiments provide further empirical evidence of the dependence of the recovery rate with SDP on the algebraic connectivity, irregularity of the graph.

**Q2-3 Extent To Which Claims Are Supported By Evidence:**

3: Good: the main claims are supported by convincing evidence (in the form of adequate experimental evaluation, proofs, (pseudo-)code, references, assumptions).

**Q2-4 Reproducibility:**

4: Excellent: key resources (e.g. proofs, code, data) are available and key details (e.g. proof sketches, experimental setup) are comprehensively described for competent researchers to confidently and easily reproduce the main results.

**Q3 Main Strengths:**

- The paper makes a non-trivial contribution to the state of the art by relaxing the missingness mechanism assumption
- Ideas and reasoning are clearly presented and the paper is well written
- Simulation settings and data are comprehensively detailed

**Q4 Main Weakness:**

1. The gain in relaxing the missingness mechanism is not completely clear in the presentation.
2. The impact of sample size is ignored in the analysis.
3. The claim that recovery rate with SDP is solely determined by the noise variance  and the graph connectivity, irregularity and spectral gap is not completely supported.

**Q5 Detailed Comments To The Authors:**

On 1. the gain that relaxing the missingness mechanism represents:
- It would be interesting to showcase examples where assuming the missingness mechanisms considered in previous literature is unreasonable and see how SDP performs in those specific cases.
- It would be interesting to compare the strength of sufficient conditions (2) to sufficient conditions when a missingness mechanism is assumed. Does it recover known particular cases of missing mechanism? Is it much more stringent than when the missingness mechanism is assumed?

On 2. The impact of sample size is ignored in the analysis:
- PCA is a data-based method applied to sample covariance matrices that typically have decreasing variance as the sample size grows. This is an important aspect for support recovery. Maybe at least mention that $\sigma$ can be regarded as functions of the sample size $n$.
- It would seem natural that the convergence rate features some quantity depending directly on the sample size. It is currently only shown as directly dependent on the dimension:  $O(log^{-1}d)$

On 3. On the claim that your empirical results “justify [your] theorem” and show that “performance of the SDP algorithm is solely determined by the properties [you] derive in [your] theorem” (implicitly solely determined by the noise variance and the graph connectivity, irregularity and spectral gap):
- Figure 5 shows SDP with completion instead of 0 imputation is not very dependent on the ratio of connectivity and irregularity. So the claim can seem bold or not so precise.
- It would be interesting to disentangle the impact of the graph quantities you single out from that of just the proportion of missing entries. Maybe by more analysis of the theoretical results or experiments.

You provide an oracle value for tuning parameter $\rho$ in Theorem 1 but then use $C_{\rho}$ to choose $\rho$ without discussing the correspondence between the two. It would be interesting to discuss whether $C_{\rho}$ recovers something similar to the oracle value in the synthetic data analysis where all problem parameters are known.

**Q9 Complying With Reviewing Instructions:**

Yes

---

> ### Author Rebuttal · Authors · 2024-04-09
>
> We thank the reviewer for the thoughtful feedback.
>
> >It would be interesting to showcase examples where assuming the missingness mechanisms considered in previous literature is unreasonable and see how SDP performs in those specific cases.
>
> One can consider the anomaly detection problem in network datasets as an example.
> Let's take an edge-weighted network dataset, where each edge's weight represents a certain type of information, such as the similarity of two nodes, traffic volume between connected locations, or social distance between individuals or groups.
> By applying sparse PCA to the weight matrix of the network, we can identify anomalous collections of individuals or nodes.
> However, since unconnected nodes lead to missingness, analysis for incomplete data must be considered.
> In this case, missingness occurs based on the network structure, not due to any random mechanisms.
>
> SDP will perform well if the missing instances exhibit favorable properties; specifically, SDP will succeed when we observe entries abundantly and evenly in the sub-matrix relevant to the true support $J$.
>
> >It would be interesting to compare the strength of sufficient conditions (2) to sufficient conditions when a missingness mechanism is assumed. Does it recover known particular cases of missing mechanism? Is it much more stringent than when the missingness mechanism is assumed?
>
> We compare our results with those of Lee et al. (2022), assuming a uniformly random missing mechanism.
> In their work, Lee et al. (2022) demonstrated under specific assumptions,
> the sufficient condition becomes $p=w\Big(\frac{1}{(\log s)^{-1}+1}\Big)$, where $p$ denotes the observation rate.
> Under the uniformly random missing scheme, the graph properties in our theorem become, with high probability,
> $\psi(G_{J,J})=\Omega(\sqrt{p(1-p)s\log s})$,
> $\phi(G_{J,J})=O(ps)$,
> $\Delta_{\max}(G_{J,J})=O(ps)$, and
> $\max(\Delta_{\max}(G_{J,J^c}), \Delta_{\max}(G_{J^c,J^c}))=O(pd)$.
> Utilizing these quantities,
> our sufficient condition is expressed as
> $p=w\Big(\frac{1}{(s^2 \log s)^{-1}+1}\Big)$
> under the same assumptions as in Lee et al. (2022).
> Our sufficient condition is more stringent than that of Lee et al. (2022),
> but it can be disregarded in the case of $s \ll d$.
>
> >The impact of sample size is ignored in the analysis.
>
> If we assume that the target matrix is $\Sigma$, and $X_i$'s are drawn from the multivariate normal distribution $N_d(0,\Sigma)$, and if we utilize the sample covariance matrix $\hat{\Sigma} = n^{-1}\sum_{i=1}^{n}X_i X_i^\top$ in the analysis, then in our theorem, $\sigma$ can be replaced by $||\Sigma||_2\cdot \frac{\log d}{\sqrt{n}}$; in this case, the convergence rate remains the same.
> We will incorporate this discussion into the paper.
>
> >Figure 5 shows SDP with completion instead of 0 imputation is not very dependent on the ratio of connectivity and irregularity. So the claim can seem bold or not so precise.
>
> Our wording was somewhat ambiguous.
> When we stated 'the performance of the SDP algorithm is solely determined by the properties we derive in Theorem 1,' we intended to refer specifically to the SDP algorithm with the zero imputation approach.
> We will make this clarification in the paper.
>
> >It would be interesting to disentangle the impact of the graph quantities you single out from that of just the proportion of missing entries. Maybe by more analysis of the theoretical results or experiments.
>
> We would like to note that the algebraic connectivity in our theorem is related to the proportion of observed entries; as more entries are observed, the algebraic connectivity tends to increase. Particularly, in situations where missingness occurs uniformly at random, the algebraic connectivity asymptotically scales with the observation rate.
> Therefore, examining the impact of the proportion of missing entries can be done by assessing the algebraic connectivity in our theorem.
>
> >It would be interesting to discuss whether $C_\rho$
> recovers something similar to the oracle value in the synthetic data analysis where all problem parameters are known.
>
> Below is a comparison between the exact recovery rate results obtained using $C_\rho$ and those obtained using an oracle value for $\rho$ in semi-synthetic data analysis of Section 5.2.
> While the results using an oracle value are slightly superior to those using $C_\rho$, they are comparable overall.
> We will add a discussion regarding this comparison.
>
> |$\frac{\psi(G_{J,J})}{\phi(G_{J,J})}$|0.1|0.3|0.5|0.7|0.9|1.1|1.3|1.5|1.7|1.9|2.1|
> |-|-|-|-|-|-|-|-|-|-|-|-|
> |$C_\rho$ | 0.79 | 0.61 | 0.61 | 0.24 | 0.34 | 0.27 | 0.22 | 0.18 | 0.08 | 0.09 | 0.06 |
> |Oracle | 0.86 | 0.67 | 0.75 | 0.34 | 0.37 | 0.38 | 0.26 | 0.23 | 0.13 | 0.17 | 0.15 |

---

### Official Review · Reviewer_FEXi · 2024-03-23

**Q2-1 Originality-Novelty:** 2
**Q2-2 Correctness-Technical Quality:** 3
**Q2-5 Clarity Of Writing:** 2

**Q1 Summary And Contributions:**

The paper aims to study a sparse PCA algorithm for incomplete and noisy data without any specific model assumption by using classical missing data approaches. The considered algorithm is based on a semidefinite relaxation of the l1-regularized PCA problem (SDP algorithm) and it can exactly recover the true support with high probability under mild conditions.

**Q2-3 Extent To Which Claims Are Supported By Evidence:**

3: Good: the main claims are supported by convincing evidence (in the form of adequate experimental evaluation, proofs, (pseudo-)code, references, assumptions).

**Q2-4 Reproducibility:**

2: Fair: key resources (e.g. proofs, code, data) are unavailable but key details (e.g. proof sketches, experimental setup) are sufficiently well-described for an expert to confidently reproduce the main results.

**Q3 Main Strengths:**

The topic is challenging and fits the UAI’s domains. The results are quite technical and the gap between the theory and the practice is well assumed and treated.

**Q4 Main Weakness:**

The comments of the main theorems seem to me unclear. Even if the theoretical results seem correct, they are hard to well understand and the comments spark some confusions. It would be interesting to see also the aspects of dimensional reduction in the procedure.

**Q5 Detailed Comments To The Authors:**

In Section 2, it would be great to discuss the matrices that satisfy the proposed identifiability hypothesis, based on the leading eignevector. I am not sure that the information is given: how the parameters of the sub-Gaussian distribution are estimated?
The ratio on the horizontal axis on the simulations is not easy to understand; is it possible to use s instead?

**Q9 Complying With Reviewing Instructions:**

Yes

---

> ### Author Rebuttal · Authors · 2024-04-09
>
> We thank the reviewer for the thoughtful feedback.
>
> > It would be interesting to see the aspects of dimensional reduction in the procedure.
>
> Sparse PCA is an approach that enforces sparsity in the PCA solution so that dimension reduction and variable selection can be simultaneously performed [Zou et al., 2006,
> Amini and Wainwright, 2008, Journée et al., 2010, Berk and
> Bertsimas, 2019, Richtárik et al., 2021].
> In this paper, we focus only on the leading principal component (dimension reduction) and aim to exactly recover the support of the true sparse leading eigenvector (variable selection.)
>
> >It would be great to discuss the matrices that satisfy the proposed identifiability hypothesis, based on the leading eigenvector.
>
> The assumption for the identifiability of the leading eigenvector is that the first and second largest eigenvalues are distinct.
> This is a standard requirement in principal component analysis. Otherwise, even with perfect observations, any weighted sum of the two leading eigenvectors could be mistakenly identified as the true leading eigenvector.
>
> >How the parameters of the sub-Gaussian distribution are estimated?
>
> It is unnecessary to estimate the noise parameter $\sigma$ in the algorithm. Although the theoretical selection of the tuning parameter $\rho$ depends on $\sigma$, we propose an AIC-based tuning procedure for $\rho$ without the need to estimate $\sigma$.
> In this sense, we would like to note that we follow common practice as done in other studies including Wainwright (2009), Journée et al. (2010), Ma (2013), Qi et al. (2013), and Chen et al. (2020).
>
> >The ratio on the horizontal axis on the simulations is not easy to understand; is it possible to use s instead?
>
> Our primary focus in this paper is to examine how graph properties (i.e., the missing pattern) influence the solvability of the sparse PCA problem.
> The ratio $\frac{\psi(G_{J,J})}{\phi(G_{J,J})}$ serves as a measure of this effect; as the ratio decreases, the sub-graph $G_{J,J}$ becomes more well-connected and has similar node degrees.
> This implies that the entries of the corresponding $|J|\times|J|$ sub-matrix are observed more abundantly and evenly.
>
> $s$ represents the intrinsic difficulty of the sparse PCA problem, which is irrelevant to the missing scheme. Therefore, the influence of $s$ is not our primary focus.

---

### Meta-Review · Area_Chair_HiTE · 2024-04-17

The paper investigates a sparse PCA algorithm for incomplete and noisy data, without specific model assumptions. The algorithm, a semidefinite relaxation of the l1-regularized PCA problem (SDP), can recover the true support with high probability under mild conditions. The study characterizes missingness patterns using a graph and demonstrates that support recovery is achievable when the graph exhibits favorable structural properties.

The reviewers seem all happy (or very happy) about the paper. The paper adds to our understanding of sparse PCA and is well written. The rebuttal from the authors was very well done and thorough - one undecided referee increased their recommendation.

The theory looks clean and it builds up on well established techniques. Empirical evidence seems to confirm that the SDP algorithm outperforms other sparse PCA approaches. Overall, the paper is very well executed but I cannot see here substantially new ideas.